# Normalized Space Alignment: A Versatile Metric for Representation Space Discrepancy Minimization

## Abstract

We introduce a manifold analysis and mapping technique for quantifying the discrepancy between two representation spaces. Normalized Space Alignment (NSA) aims to compare pairwise distances between two point clouds. Uniquely positioned as both an analytical tool and a differentiable loss function, NSA provides a robust means of comparing and aligning representations across different layers and models. We show that our technique acts as a pseudometric, satisfies the properties of a similarity metric, and is continuous and differentiable. NSA can serve as an auxiliary loss function in neural networks to preserve representation structure. NSA is not only computationally efficient but it also can effectively approximate the global structural discrepancy during mini-batching, facilitating its use in large neural network training paradigms. Output representations generated by an NSA enhanced neural network are shown to be effective in multiple downstream tasks. Its versatility extends to robustness analysis and various neural network training and representation learning applications, highlighting its wide applicability and potential to enhance the performance of neural networks.

## 1 Introduction

Deep learning and representation learning have rapidly emerged as cornerstones of modern artificial intelligence, revolutionizing how machines interpret complex data. At the heart of this evolution is the ability of deep learning models to learn efficient representations from vast amounts of unstructured data, transforming it into a format where patterns become discernible and actionable. This capability has unlocked unprecedented achievements in fields ranging from computer vision to natural language processing. As these models delve deeper into learning intricate structures and relationships within data, the necessity for advanced metrics that can accurately assess and preserve the integrity of these learned representations becomes increasingly evident. The ability to preserve embedding structure would enhance the performance, interpretability and robustness (Finn et al., 2017) of these models.

Although existing methods have been pivotal in achieving state-of-the-art results in several deep learning domains, they do not attempt to minimize the global structural discrepancy between spaces, instead focusing on improving relative positions of similar embeddings (Chen et al., 2020; Schroff et al., 2015). How to quantify global structural similarity of representations is an unsolved problem. Early proposed measures were based on variants of "Canonical Correlation Analysis (CCA)" (Morcos et al., 2018; Raghu et al., 2017), and "Centered Kernel Alignment (CKA)" (Kornblith et al., 2019). Recently Barannikov et al. (2022) proposed methods for comparing two data representations using "Representation Topology Divergence (RTD)" that measures the dissimilarity in multi-scale topology between two point clouds of equal size with a one-to-one correspondence between points.

There exists a compelling need for a novel approach that bridges the gap between two distinct yet essential attributes: computational complexity and differentiability. While CKA offer an efficient means of assessing similarity, it has been proven to be lacking on several fronts (Davari et al., 2023; Williams et al., 2021; Ding et al., 2021). On the other hand, metrics like RTD provide differentiability but at the expense of computational complexity, making them impractical for large-scale datasets. Addressing this need for a similarity metric that marries the computational efficiency of CKA with

the differentiability of RTD is crucial. Such a metric not only empowers practitioners to navigate high-dimensional spaces with ease but also facilitates the seamless integration of similarity-based techniques into gradient-based optimization pipelines, thereby unlocking new frontiers in machine learning and data analysis.

We present a distance preserving similarity metric called "Normalized Space Alignment (NSA)." NSA measures the work done to align all the points in one representation space $P$ to another $Q$ when both spaces are normalized to lie in unit space. Both representations can lie in different ambient spaces as long as there is a one to one mapping between the data points. The work done is measured as the difference between the normalized distance of a point to all other points in the two representations. When computed for all the points in the space, the mean of the absolute work done defines the NSA between $P$ and $Q$.

The efficiency and representativeness of metrics are crucial for practical applicability. A viable metric must be computationally efficient to facilitate its use in large-scale datasets and real-time training scenarios. Equally important is the metric's ability to reflect global structural discrepancies, even when applied in mini-batching contexts. Metrics that scale efficiently with data size and can approximate the global discrepancy through batched analysis prove indispensable in neural network training, where they enable accurate minimization of structural discrepancies between representation spaces. We prove that NSA not only satisfies these conditions, but is currently the only metric that manages to do so. Our contributions are summarized below:

- NSA is proposed and established as pseudometric and a similarity index. NSA's quadratic computational complexity (in the number of points) is much better than some of the existing measures (e.g., cubic in the number of simplices for RTD). NSA is shown to be continuous and differentiable, facilitating its use as a loss function in neural networks. The calculation of NSA over a subset of the data is shown to be representative of the global NSA value.

- NSA's performance as a loss function is evaluated in Section 4. NSA is used as a supplementary loss in autoencoders for dimensionality reduction. Results on several datasets show that NSA can better preserve structural characteristics of the original data compared to its competitors. Further downstream task analyses shows that the latent embeddings obtained from NSA better preserve semantic relationships between the data. NSA's ability to map geodesic distances when reducing representations to their manifold dimension is also explored.

- NSA's performance is evaluated on several popular empirical tests, designed to determine if a similarity index is performing as expected (Section 5). Derived from the experiments published by Kornblith et al. (2019), NSA is shown to achieve exceptional performance on the similarity index sanity test, show a strong correlation to test accuracy of data, predict convergence and provide insights on similarity across architectures and downstream tasks.

- NSA can capture structural discrepancies that result from adversarial attacks (Section 6). NSA's performance on global poisoning and evasion attacks on several GNN architectures is evaluated. NSA shows a high correlation with misclassification rate across different degrees of perturbation. Nodewise NSA can also provide insights on node vulnerability and the inner workings of popular defense methods. Furthermore, it can also rank different architectures. Simple tests with NSA can provide insights on par with other works that review and investigate the robustness of GNNs (Mujkanovic et al., 2022; Jin et al., 2020a).

## 2 RELATED WORK

**Representation Similarity Measures:** Centered Kernel Alignment (CKA) and Representation Topology Divergence (RTD) are two notable representation similarity measures in neural network analysis. CKA, utilizing a Representational Similarity Matrix (RSM) approach with mean-centered representations and a linear kernel, focuses on measuring similarity invariant to scaling and rotation, though it lacks sensitivity to certain representational changes (Ding et al., 2021) and does not satisfy the triangle inequality (Williams et al., 2021). RTD, on the other hand, approximates representation manifolds using simplices to compare topological feature discrepancies between representations, offering better correlation with model prediction disagreements than CKA. However, RTD's high

computational complexity and limited practicality for larger sample sizes constrain its wider application.

**Dimensionality Reduction:** Recent research in dimensionality reduction (Trofimov et al., 2023; Moor et al., 2020; Rudolph et al., 2019) addresses the challenges of high-dimensional data. Classical methods like PCA and MDS, as well as local structure-preserving approaches like t-SNE (van der Maaten & Hinton, 2008), UMAP (McInnes et al., 2018), and PaCMAP (Wang et al., 2021), remain popular. However, these methods are less practical for large real-world datasets due to high computational demands. Autoencoders (Hinton & Salakhutdinov, 2006) and variational autoencoders (Kingma & Welling, 2022) offer interpretable low-dimensional representations but may not preserve the initial data's topology. Topological autoencoders (Moor et al., 2020) introduced an additional loss term to maintain topological characteristics in latent representations. Similarly, RTD-AE (Trofimov et al., 2023) enhanced autoencoders with their topology-preserving metric RTD and outperformed TopoAE and classical techniques, establishing itself as the state-of-the-art in dimensionality reduction.

## 3 NORMALIZED SPACE ALIGNMENT

NSA comes under a category of similarity metrics called Representational Similarity Matrix-Based (RSM) Measures (Klabunde et al., 2023). Given a representation $\mathbf{R} : N \times D$, all RSM based measures generate a matrix of instance wise similarities $D \in \mathbf{R}^{N \times N}$ where $D_{i,j} := d(R_i, R_j)$. Now for two representations $\mathbf{R}$ and $\mathbf{R}'$, we obtain two RSMs which we can operate upon to compute a similarity measure. Kornblith et al. (2019); Székely et al. (2007); Kriegeskorte et al. (2008); Chen (2022) are popular examples of RSM based similarity measures.

### 3.1 DEFINITION

The following formula is used to compute the NSA between two point clouds $X = \{\vec{x_1}, \ldots, \vec{x_N}\}$ and $Y = \{\vec{y_1}, \ldots, \vec{y_N}\}$. $d(\cdot, \cdot)$ denotes a differentiable distance measure between two vectors.

$$\mathsf{NSA}(X, Y, i) = \frac{1}{N} \sum_{1 \leq j \leq N} \left| \frac{d(\vec{x_i}, \vec{x_j})}{\max_{\vec{x} \in X} d(\vec{x}, \vec{0})} - \frac{d(\vec{y_i}, \vec{y_j})}{\max_{\vec{y} \in Y} d(\vec{y}, \vec{0})} \right|.^1$$

$$\mathsf{NSA}(X, Y) = \frac{1}{N} \sum_{1 \leq i \leq N} \mathsf{NSA}(X, Y, i).$$

Throughout the paper, we use $d(\cdot, \cdot)$ to denote the euclidean distance (unless specified otherwise).

Essentially, we compute the differences of normalized distances (L1-norm) of a point to all other points in the two representations, and then take the average of all these differences.

### 3.2 NSA AS A PSEUDOMETRIC

We show that NSA is a psuedometric by proving the necessary properties in the Appendix: • $\mathsf{NSA}(X, X) = 0$ (lemma 1), • Symmetry (lemma 2), • Non-negativity (lemma 3), • Triangle Inequality (lemma 4).

### 3.3 NSA AS A SIMILARITY METRIC

We adopt the necessary conditions of Invariance to Isotropic Scaling, Invariance to Orthogonal Transformation, and (Not) Invariance to Invertible Linear Transformation (ILT) as proposed by Kornblith et al. (2019) for a similarity metric. We establish these condition in Appendix B: lemma 7), lemma 9, and B.3. These conditions ensure that the similarity metric would be unaffected by rotation or rescaling of the representation space. Kornblith et al. (2019) proved that Invariance to ILT for a similarity metric gives the same result for any representation having width greater than or equal to

---

$^1\vec{0}$ denotes a vector (of appropriate size) containing all 0s.

the dataset size. They also discuss other scenarios wherein Invariance to ILT would be detrimental to similarity indices.

## 3.4 COMPLEXITY ANALYSIS

NSA's computational complexity is given by $O(N^2 D)$ where $N$ is the number of datapoints and $D$ is $\max(D(R), D(R'))$ where $R$ and $R'$ are the two representation spaces and $D(\cdot)$ is the dimensionality of the space. In practice, NSA's computation is rapid as we use torch.cdist to compute pairwise distances on the GPU. CKA has similar complexity to NSA. While RTD also starts off with $O(N^2 D)$ operations to generate pairwise distances, the barcode computation is cubic in the number of simplices involved which could be significantly more than the number of points. Running times of NSA-AE are given in Table 3. NSA-AE is several times faster than RTD-AE and it can run on much larger batch sizes.

## 3.5 IMPROVING ROBUSTNESS OF NSA

Formally, NSA is normalized by the point that is furthest away from the origin for the representation space. In practice, NSA tends to yield more robust results and exhibits reduced susceptibility to outliers when distances are normalized by scaling relative to a quantile of the distances measured from the origin (e.g., 0.98 quantile). This quantile-based approach ensures that the distances are adjusted proportionally, taking into account the spread of values among the points with respect to their distances from the origin. It also ensures that any outliers do not affect the rescaling of the point cloud to the unit space.

## 4 NSA AS A LOSS FUNCTION

The viability of NSA as a loss function is established by demonstrating its non-negativity (lemma 3), nullity (lemma 1) and continuity, and by developing a differentiation scheme. Proofs for all 4 properties are presented in the Appendix C.We also demonstrate that the expectation of NSA over a minibatch is equal to the NSA of the whole dataset, cementing its feasibility as a loss function in Appendix D.

### 4.1 DEFINING NSA-AE: AUTOENCODER USING NSA

To evaluate the efficacy of NSA as a structural discrepancy minimization metric, we take inspiration from TopoAE and RTD-AE, and use NSA as loss function in autoencoders for dimensionality reduction, thus defining NSA-AE. A normal autoencoder aims to minimize the MSE loss between the original $X$ and reconstructed embedding $\hat{X}$. We add NSA Loss as an additional loss term that aims to minimize the discrepancy in representation structure between the original embedding space $X$ and the latent embedding space $Z$. The autoencoder is built as a compression autoencoder where the encoder attempts to reduce the original data to a latent dimension and the decoder attempts to reconstruct the original embedding. Since NSA can be used in autoencoders with mini-batch training, NSA-AE runs almost as fast as a regular autoencoder. We compare the performance of NSA-AE against PCA, UMAP, a regular autoencoder, TopoAE and RTD-AE on four real world datasets.

The performance of NSA-AE is evaluated on the structural and topological similarity between the input data $X$ and the latent data $Z$. In order to evaluate the performance, we use: (1) linear correlation of pairwise distances, (2) triplet distance ranking accuracy (Wang et al., 2021), (3) RTD, (4) triplet distance ranking accuracy between cluster centers, and (5) NSA. As seen in Table 1, NSA-AE has better correlation between the original data and the latent data compared to the other models. NSA-AE outperforms the other approaches on all metrics but RTD, where it ranks just below RTD-AE. NSA-AE achieves MSE and running times similar to a normal autoencoder while previous works are several times slower, as shown in Table 3 in the Appendix.

### 4.2 DOWNSTREAM TASK ANALYSIS

We demonstrate that NSA-AE effectively preserves the structural integrity between the original data and the latent space. However, it is imperative to note that such preservation does not inherently

| Dataset | Method | Quality measure | | | | |
|---------|--------|------|------|------|------|------|
| | | L. C. | T. A. | RTD | T. A. C.C | NSA |
| MNIST | PCA | 0.910 | $0.871 \pm 0.008$ | $6.69 \pm 0.21$ | $\mathbf{0.986 \pm 0.117}$ | $0.0817 \pm 0.0025$ |
| | UMAP | 0.424 | $0.620 \pm 0.013$ | $18.06 \pm 0.48$ | $0.824 \pm 0.381$ | $0.2305 \pm 0.0031$ |
| | AE | 0.801 | $0.778 \pm 0.007$ | $7.47 \pm 0.20$ | $0.828 \pm 0.377$ | $0.0571 \pm 0.0011$ |
| | TopoAE | 0.765 | $0.771 \pm 0.010$ | $6.16 \pm 0.23$ | $0.886 \pm 0.318$ | $0.0477 \pm 0.0011$ |
| | RTD-AE | 0.837 | $0.811 \pm 0.004$ | $\mathbf{4.26 \pm 0.14}$ | $0.842 \pm 0.365$ | $0.1694 \pm 0.0024$ |
| | NSA-AE | $\mathbf{0.942}$ | $\mathbf{0.885 \pm 0.006}$ | $5.44 \pm 0.12$ | $0.944 \pm 0.230$ | $\mathbf{0.0198 \pm 0.0001}$ |
| F-MNIST | PCA | 0.978 | $0.951 \pm 0.006$ | $5.91 \pm 0.12$ | $\mathbf{1.000 \pm 0.0}$ | $0.1722 \pm 0.0038$ |
| | UMAP | 0.592 | $0.734 \pm 0.012$ | $12.16 \pm 0.39$ | $0.916 \pm 0.277$ | $0.1420 \pm 0.0011$ |
| | AE | 0.872 | $0.850 \pm 0.008$ | $5.60 \pm 0.21$ | $0.926 \pm 0.262$ | $0.0527 \pm 0.0028$ |
| | TopoAE | 0.875 | $0.854 \pm 0.009$ | $4.27 \pm 0.15$ | $0.946 \pm 0.226$ | $0.111 \pm 0.0027$ |
| | RTD-AE | 0.949 | $0.902 \pm 0.004$ | $\mathbf{3.05 \pm 0.12}$ | $0.972 \pm 0.165$ | $0.0349 \pm 0.0015$ |
| | NSA-AE | $\mathbf{0.987}$ | $\mathbf{0.952 \pm 0.002}$ | $4.11 \pm 0.21$ | $0.992 \pm 0.089$ | $\mathbf{0.0091 \pm 0.0001}$ |
| CIFAR-10 | PCA | 0.972 | $0.926 \pm 0.009$ | $4.99 \pm 0.16$ | $\mathbf{0.994 \pm 0.077}$ | $0.1809 \pm 0.0046$ |
| | UMAP | 0.756 | $0.786 \pm 0.010$ | $12.21 \pm 0.22$ | $0.956 \pm 0.205$ | $0.1316 \pm 0.0026$ |
| | AE | 0.834 | $0.836 \pm 0.006$ | $4.07 \pm 0.28$ | $0.920 \pm 0.271$ | $0.0616 \pm 0.0019$ |
| | TopoAE | 0.889 | $0.854 \pm 0.007$ | $3.89 \pm 0.11$ | $0.942 \pm 0.234$ | $0.0625 \pm 0.0014$ |
| | RTD-AE | 0.971 | $0.922 \pm 0.002$ | $\mathbf{2.95 \pm 0.08}$ | $0.976 \pm 0.153$ | $0.0113 \pm 0.0003$ |
| | NSA-AE | $\mathbf{0.985}$ | $\mathbf{0.936 \pm 0.004}$ | $3.07 \pm 0.11$ | $0.984 \pm 0.125$ | $\mathbf{0.0077 \pm 0.0001}$ |
| COIL-20 | PCA | $\mathbf{0.966}$ | $0.932 \pm 0.005$ | $6.49 \pm 0.23$ | $\mathbf{0.992 \pm 0.090}$ | $0.2204 \pm 0.0$ |
| | UMAP | 0.274 | $0.567 \pm 0.016$ | $15.50 \pm 0.67$ | $0.669 \pm 0.471$ | $0.1104 \pm 0.0$ |
| | AE | 0.850 | $0.836 \pm 0.008$ | $9.57 \pm 0.27$ | $0.889 \pm 0.314$ | $0.0758 \pm 0.0$ |
| | TopoAE | 0.804 | $0.805 \pm 0.011$ | $7.33 \pm 0.21$ | $0.885 \pm 0.319$ | $0.0676 \pm 0.0$ |
| | RTD-AE | 0.908 | $0.871 \pm 0.005$ | $\mathbf{5.89 \pm 0.10}$ | $0.891 \pm 0.311$ | $0.0523 \pm 0.0$ |
| | NSA-AE | 0.955 | $\mathbf{0.919 \pm 0.004}$ | $7.46 \pm 0.23$ | $0.939 \pm 0.240$ | $\mathbf{0.0157 \pm 0.0}$ |

Table 1: Autoencoder results. NSA-AE outperforms or almost matches all other approaches on all the evaluation metrics. RTD-AE, which explicitly minimizes on RTD has a slightly lower RTD value while PCA has marginally higher Triplet Ranking Accuracy on Cluster Centers.

ensure the utility of the resultant latent embeddings. Metrics like NSA, which focus on global structure preservation, excel in scenarios where the preservation of semantic relationships between data points holds paramount importance.

Link prediction is an ideal task to demonstrate NSA's structure preserving ability. Successful link prediction mandates a global consistency within the embedding space, necessitating that nodes with likely connections are proximate while dissimilar nodes are distant. Any form of clustering or localized alterations to the representation structure can disrupt node interrelationships. In our study, we employ a Graph Convolutional Network (GCN) to train link prediction models across four distinct graph datasets. The original GCN embedding space encompasses 256 dimensions. This space is subsequently processed through NSA-AE, resulting in a reduced-dimensional representation. Notably, the latent embeddings produced by NSA-AE exhibit superior performance when compared to both a conventional autoencoder and RTD-AE as shown in Table 2. Additionally, we present results showcasing the effectiveness of semantic textual similarity matching using Word2Vec embeddings in the Appendix M.

## 4.3 APPROXIMATING THE MANIFOLD OF A REPRESENTATION SPACE

Incorporating a structure-preserving metric with Mean Squared Error (MSE) in an autoencoder, NSA minimizes the Euclidean distance between points in a representation space to reduce data dimensionality while maintaining structure. High-dimensional data typically resides in a lower-dimensional manifold, where the Euclidean distance approximates the geodesic distance. This property is exploited in NSA to interpret data with misleading ambient distances.

Consequently, when armed with knowledge pertaining to the manifold dimension of the input data and the geodesic distances between data points, NSA-AE can be trained to minimize the Euclidean distance between these points. This optimization process effectively results in the reduction of

| Method | Latent Dim | Dataset | | | |
|---|---|---|---|---|---|
| | | Amazon Comp | Cora | Citeseer | Pubmed |
| AE | 256* | 97.56 | 97.89 | 97.25 | 97.61 |
| | 128 | 50.478 | 50.0 | 50.0 | 51.17 |
| | 64 | 50.56 | 50.0 | 50.0 | 50.03 |
| | 32 | 59.63 | 49.84 | 49.80 | 52.78 |
| RTD-AE | 256* | 95.81 | 97.23 | 97.53 | 97.58 |
| | 128 | 94.63 | 86.52 | 90.54 | 91.80 |
| | 64 | 91.63 | 82.38 | 89.23 | **97.26** |
| | 32 | 88.34 | 83.13 | 57.18 | 87.05 |
| NSA-AE | 256* | 96.12 | 96.31 | 98.77 | 97.57 |
| | 128 | **95.74** | **96.04** | **97.18** | **97.31** |
| | 64 | **95.90** | **95.54** | **96.85** | 96.40 |
| | 32 | **96.01** | **95.37** | **94.25** | **97.53** |

Table 2: Downstream task analysis with Link Prediction. The output embeddings from a GCN trained on link prediction are passed through all 3 autoencoder architectures. Latent embeddings are obtained at various dimensions and ROC-AUC scores are calculated on the latent embeddings. The first row for every architecture shows the ROC-AUC score for the reconstructed embeddings, proving that none of the architectures compensate on reconstruction accuracy and work well with MSE Loss. The other rows show the scores for the latent embeddings. NSA-AE outperforms both a regular autoencoder and RTD-AE across all datasets and all latent dimensions.

the input data to its intrinsic manifold shape. We show the results of mnimizing for euclidean distance and geodesic distance for the Swiss Roll Dataset in Figure 1. Conversely, in cases where the manifold dimension of the input data remains unknown, the latent dimension at which NSA minimization becomes optimal inherently corresponds to the manifold dimension of the data. This observation underscores the practical significance of understanding the manifold dimension when leveraging NSA for dimensionality reduction within an autoencoder framework.

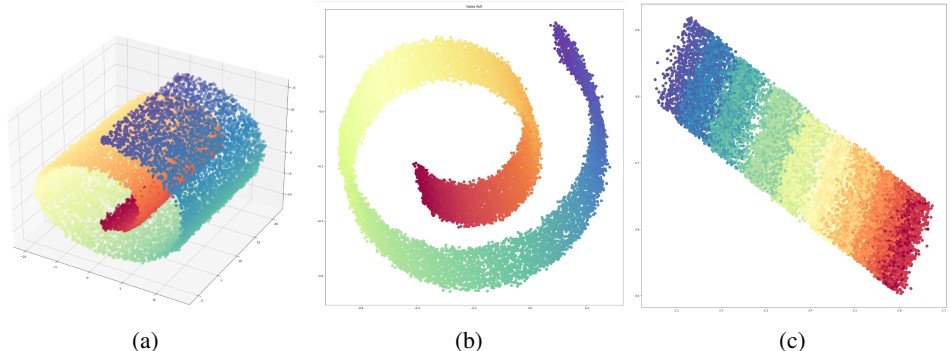

(a)  (b)  (c)

Figure 1: Results of NSA-AE on the Swiss Roll Dataset. (a) The original Swiss Roll Dataset in 3D. (b) Result of dimensionality reduction to 2D using NSA-AE when minimizing euclidean distance. (c) Result of dimensionality reduction to 2D using NSA-AE when minimizing geodesic distance

## 5 EMPIRICAL ANALYSIS OF NSA

Empirical Analysis of intermediate representations with similarity measures can help uncover the intricate workings of various neural network architectures. Historically, much of this empirical scrutiny has been directed towards Convolutional Neural Networks (CNNs) Kornblith et al. (2019); Barannikov et al. (2022) leaving analysis of other domains untouched. Graph Neural Networks (GNNs), in particular, present a unique and relatively unexplored territory. Modeling graphs has been particularly challenging because of impossibility results on isometric embeddings (no matter how many target dimensions) and bounds on distortions of their embedding into Euclidean

spaces Bourgain (1985); Linial et al. (1994). GNNs, with their inherent complexity, provide a robust testing ground for NSA. NSA is designed to measure 'dissimilarity' between two embedding spaces as long as a one to one mapping exists between the points in both embedding spaces, regardless of their originating architecture or dimensionality.

## 5.1 COMPARING DIFFERENT INITIALIZATIONS

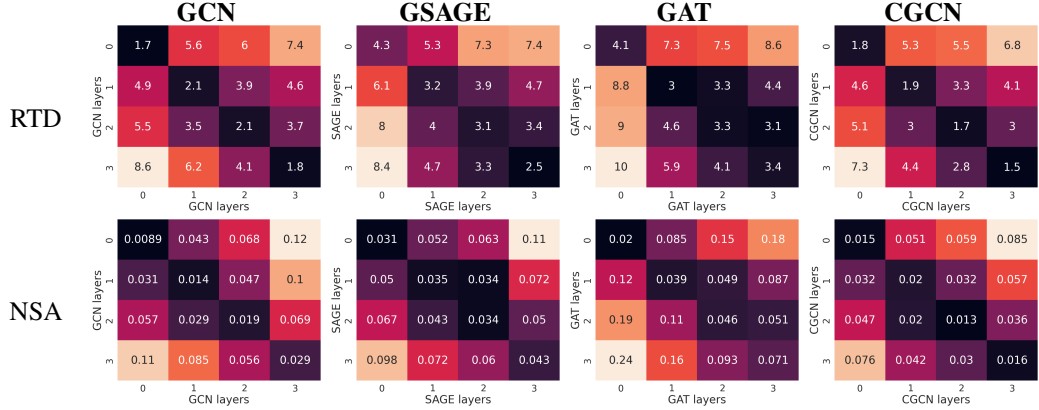

Figure 2: Sanity Tests for Node Classification (Amazon Computers Dataset). The heatmaps show layer-wise dissimilarity values for two different initialization of the same dataset on four different GNN architectures. The first row shows the RTD values and the second row shows the NSA values. NSA shows a stronger layer-wise correlation. Results for CKA' are in Appendix Q

We conduct a basic sanity test for evaluating similarity metrics as suggested by Kornblith et al. (2019). When two networks that are structurally identical but trained from different initial conditions are compared, we expect that, for each layer's intermediate representation, the most similar representation in the other model should be the corresponding layer that matches in structure. We perform the tests when the models are trained on Node Classification on the Amazon Computers dataset (Shchur et al., 2019) in Figure 2. At the end of training, the intermediate embeddings of the input data at each layer are extracted and utilized to compute the similarity between two layers. Barannikov et al. (2022) and Chen (2022) show that similarity indices can capture the structure of an embedding space with just a subset of the entire data. We use a sample of 4000-9000 data points to compute the NSA between two representation spaces and the recommended sample size of 400 points for RTD.

We utilize four different GNN architectures; Graph Convolution Networks (Kipf & Welling, 2017), GraphSAGE (Hamilton et al., 2017), Graph Attention Networks (Veličković et al., 2018) and ClusterGCN (Chiang et al., 2019) for our experiments. We also showcase the results of CKA' and RTD to empirically demonstrate that NSA is more nuanced than RTD and CKA', and is capable of identifying patterns that are expected based on the design of the architectures. Both NSA and RTD compare dissimilarity while CKA compares similarity. For ease of comparison between the metrics, throughout this paper we use $1 - \mathsf{CKA}$ to show the performance of CKA and label it $\mathsf{CKA}'$. For further details on the architectures, model setup and hyperparameters, please refer to Appendix N.

## 5.2 CONVERGENCE TESTS

We examine epoch-wise convergence by comparing representations between the current and final epochs. Figure 3 illustrates our results for GAT and CGCN when trained on the Amazon Dataset on node classification. We observe that NSA's convergence corresponds with the test accuracy convergence of the model. This shows that NSA can be used as an early stopping measure during neural network training, stopping the weight updates once the structure of your intermediate representations stabilize. We show results for other architectures on node classification and link prediction with GCN and GraphSAGE in Figure 17.

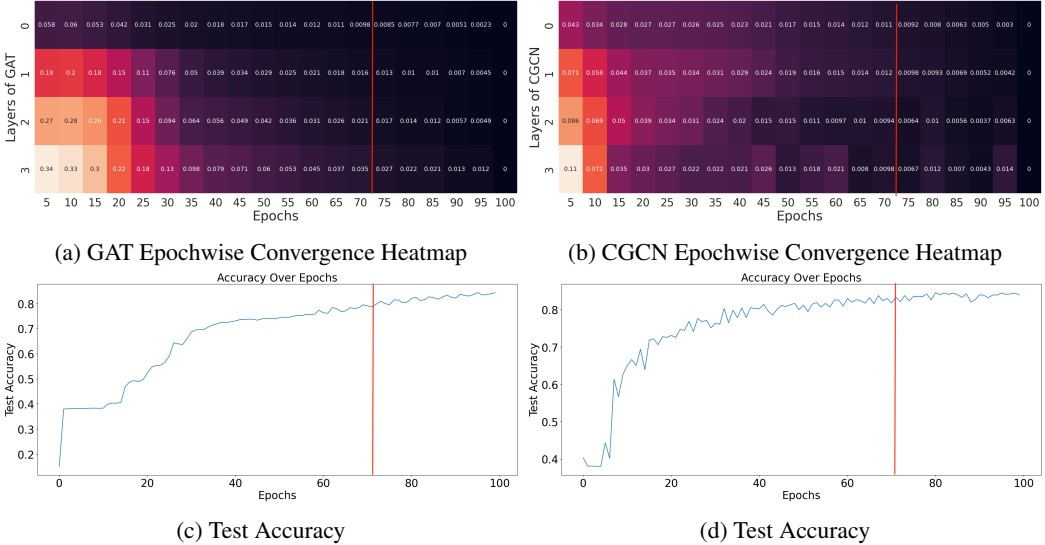

(a) GAT Epochwise Convergence Heatmap      (b) CGCN Epochwise Convergence Heatmap

(c) Test Accuracy                (d) Test Accuracy

Figure 3: Convergence Tests for Node Classification on the Amazon Computers Dataset. The representation for each layer is compared epochwise against that layer's final epoch representation. NSA's convergence corresponds strongly with the test accuracy convergence for both models. The red line shows the point at which the model's intermediate representations converge

# 6 ANALYZING ADVERSARIAL ATTACKS WITH NSA

Addressing the issue of adversarial attacks in machine learning models, particularly in the context of Graph Neural Networks (GNNs), is of paramount importance. Adversarial attacks pose a significant threat in various domains, including social networks, recommendation systems, and cybersecurity, where GNNs are extensively employed. To fortify GNNs against adversarial attacks, the incorporation of a structure-preserving minimization term in the training process is a promising approach. Such a term enforces the preservation of key structural characteristics within the data, reducing the model's susceptibility to perturbations. However, the efficacy of this approach greatly hinges on the availability of a similarity metric capable of discerning perturbations in the representation space.

In this experimental study, we subject five distinct Graph Neural Network (GNN) architectures to both poisoning and evasion adversarial attacks using projected gradient descent (Xu et al., 2019). We use the regular GCN along with four robust GNN variants; SVD-GCN (Entezari et al., 2020), GNNGuard (Zhang & Zitnik, 2020), GRAND (Chamberlain et al., 2021) and ProGNN (Jin et al., 2020b). To assess the vulnerability of these architectures, we manipulated the initial adjacency matrices by introducing perturbations ranging from 5% to 25%. The objective was to gauge the impact of these perturbations on the misclassification rates of the GNN models and to see if NSA shows a strong correlation to the misclassification rates over different perturbation rates.

NSA was computed at each stage by comparing the clean graph's output representations with those from the perturbed graph. Our experiments revealed that NSA's variation over different perturbation rates mirrors the misclassification trends of various GNN architectures during poisoning attacks and NSA's ranking is in line with previous works (Jin et al., 2020a; Mujkanovic et al., 2022). However, in evasion attacks, SVD-GCN stood out with its NSA scores significantly deviating from its misclassification rates. To elucidate this anomaly, Figure 5 examines the boundary nodes of different GNN architectures. Post-attack, SVD-GCN showed a substantial increase in boundary nodes—defined by low classification confidence and proximity to the decision boundary—accompanied by a general decline in classification confidence. This surge, coupled with SVD-GCN's highest pointwise NSA values among all tested architectures, potentially accounts for its high NSA scores and heightened vulnerability. It also this vulnerability in structure, that is exploited by Mujkanovic et al. (2022) with adaptive adversarial attacks, to cause a catastrophic failure in SVD-GCN. This detailed analysis underpins our assertion that NSA, through simple empirical analysis, can unveil concealed weaknesses in defense methods not immediately apparent from conventional metrics.

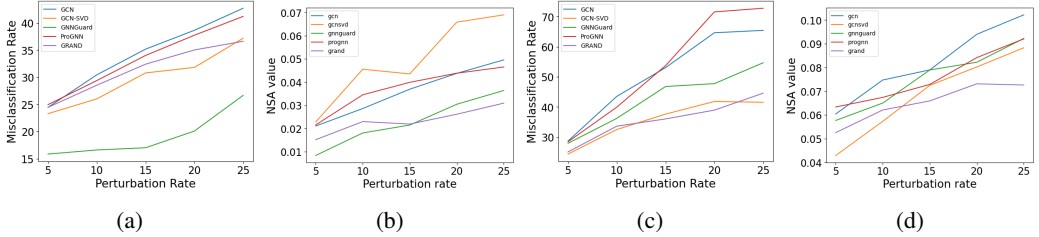

(a)       (b)       (c)       (d)

Figure 4: Robustness tests with NSA. (a) Variation of Misclassification Rate against Data Perturbation Rate for GNN architectures under global evasion attack. (b) NSA against perturbation rate for GNN architectures under global evasion attack. (c) Variation of Misclassification Rate against Data Perturbation Rate for GNN architectures under global poisoning attack. (b) NSA against perturbation rate for GNN architectures under global poisoning attack

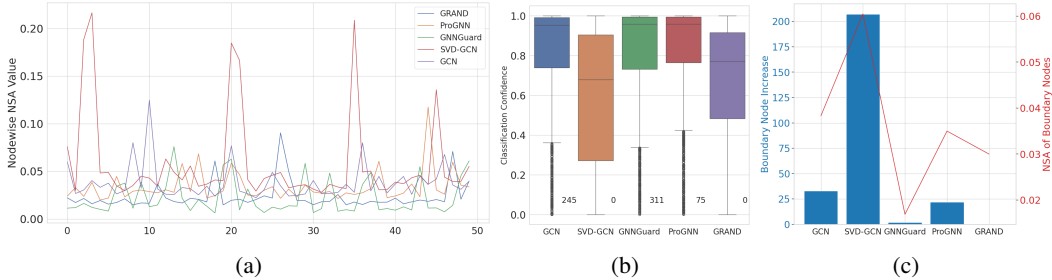

(a)       (b)       (c)

Figure 5: Analyzing node vulnerability with NSA. (a) Nodewise NSA of the 50 nodes with the greatest decline in Classification Confidence. SVD-GCN has the highest nodewise NSA variations. (b) Classification Confidence of all the nodes after an evasion attack. (c) Increase in number of boundary nodes for each model post attack and its correlation with the NSA of the boundary nodes.

## 7 DISCUSSION AND CONCLUSION

In conclusion, we have demonstrated that the proposed measure of NSA is simple, efficient, and useful across many aspects of analysis and synthesis of representation spaces: robustness across initializations, convergence across epochs, autoencoders, adversarial attacks, and effects of downstream tasks across architectures. Its computational efficiency and the ability to converge to its global value in mini batches should allow it to be useful in many applications and generalizations where scalability is desired.

One possibility for future research would be to combine NSA with measures of intrinsic dimensionality (Camastra & Staiano, 2016; Campadelli et al., 2015). This is motivated by the observation that real-world data presented in a high-dimensional space usually lies along a manifold of much lower dimension (Goodfellow et al., 2016). A challenge here would be ensuring that the resulting measure does not add much of a computational overhead to NSA.

Empirical findings by Kornblith et al. (2019) have indicated minimal improvement with the RBF variant of the Centered Kernel Alignment (CKA) over the linear kernel CKA. However, it is crucial to recognize that in high-dimensional spaces, the Euclidean distance metric may not be particularly effective, as highlighted by theoretical works such as Bellman's "The Curse of Dimensionality" (Bellman, 1961). Despite this, Euclidean distance remains prevalent in various domains, including contrastive (Chen et al., 2020) or triplet losses (Schroff et al., 2015), style transfer (Johnson et al., 2016) and similarity indices, often yielding successful empirical results. Our future investigations intend to explore alternative distance measures, potentially considering non-linear options such as geodesic distance and assessing their viability.

Finally, on the question of structural similarity and functional similarity, Davari et al. (2023) highlight CKA's susceptibility to subset translations and situations where CKA changes while functional behavior remains consistent. While NSA is more tuned to structural similarity, we plan to carry out a similar analysis in the future.

## 8 REPRODUCIBILITY STATEMENT

Our code is anonymously available at https://anonymous.4open.science/r/NSA. The code includes notebooks with instructions to reproduce all the experiments presented in this paper. To reproduce Table 1 and Table 3 you will access the NSA_AE folder. The hyperparameter setup to reproduce the autoencoder results are in Table 4. To reproduce all the heatmaps in the paper, you will need to access the GNN_analysis folder. The hyperparameter setup to reproduce these results is given in Table 6. To reproduce the results in Section 6, you will access the Adversarial Analysis folder. No hyperparameter setup is necessary to run the notebooks in this section.

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

## A    PROOFS FOR NSA AS A PSEUDOMETRIC

### A.1    NSA$(X, X) = 0$

**Lemma 1** (Identity). *Let $X$ be a point cloud over some space, then* NSA$(X, X) = 0$.

*Proof.* Let $X = \{x_1, \dots, x_N\}$. Let $\max_{x \in X}(d(x, 0)) = D$. Then

$$NSA(X, X) = \frac{1}{N^2} \sum_{1 \leq i,j \leq N} \left| \frac{d(x_i, x_j)}{D} - \frac{d(x_i, x_j)}{D} \right|.$$

Simplifying, we get

$$NSA(X, X) = \frac{1}{N^2 D} \sum_{1 \leq i,j \leq N} |d(x_i, x_j) - d(x_i, x_j)| = 0.$$

$\square$

**Lemma 2** (Symmetry). *Let $X, Y$ be two point clouds of the same size, then* NSA$(X, Y) =$ NSA$(Y, X)$.

*Proof.* Let $X = \{x_1, \dots, x_N\}$ and $Y = \{y_1, \dots, y_N\}$. Let $\max_{x \in X}(d(x, 0)) = D_X$ and $\max_{y \in Y}(d(y, 0)) = D_Y$. Then

$$\mathsf{NSA}(X, Y) = \frac{1}{N^2} \sum_{1 \leq i,j \leq N} \left| \frac{d(x_i, x_j)}{D_X} - \frac{d(y_i, y_j)}{D_Y} \right|.$$

Since $|a - b| = |b - a|$,

$$\mathsf{NSA}(X, Y) = \frac{1}{N^2} \sum_{1 \leq i,j \leq N} \left| \frac{d(y_i, y_j)}{D_Y} - \frac{d(x_i, x_j)}{D_X} \right|.$$

Hence, NSA$(X, Y) =$ NSA$(Y, X)$. $\square$

**Lemma 3** (Non-negativity). *Let $X, Y$ be two point clouds of the same size, then* NSA$(X, Y) \geq 0$.

*Proof.* Let $X = \{x_1, \dots, x_N\}$ and $Y = \{y_1, \dots, y_N\}$. Let $\max_{x \in X}(d(x, 0)) = D_X$ and $\max_{y \in Y}(d(y, 0)) = D_Y$. Then

$$\mathsf{NSA}(X, Y) = \frac{1}{N^2} \sum_{1 \leq i,j \leq N} \left| \frac{d(x_i, x_j)}{D_X} - \frac{d(y_i, y_j)}{D_Y} \right|.$$

Since, for all $i, j$,

$$\left| \frac{d(x_i, x_j)}{D_X} - \frac{d(y_i, y_j)}{D_Y} \right| \geq 0,$$

we get NSA$(X, Y) \geq 0$. $\square$

**Lemma 4** (Triangle inequality). *Let $X, Y$ and $Z$ be three point clouds of the same size, then* NSA$(X, Z) \leq$ NSA$(X, Y) +$ NSA$(Y, Z)$.

*Proof.* Let $X = \{x_1, \dots, x_N\}$, $Y = \{y_1, \dots, y_N\}$ and $Z = \{z_1, \dots, z_N\}$. Let $\max_{x \in X}(d(x, 0)) = D_X$, $\max_{y \in Y}(d(y, 0)) = D_Y$ and $\max_{z \in Z}(d(z, 0)) = D_Z$. Then

$$\mathsf{NSA}(X, Z) = \frac{1}{N^2} \sum_{1 \leq i,j \leq N} \left| \frac{d(x_i, x_j)}{D_X} - \frac{d(z_i, z_j)}{D_Z} \right|.$$

For each $i, j$, add and subtract $\frac{d(y_i, y_j)}{D_Y}$, then

$$\mathsf{NSA}(X, Z) = \frac{1}{N^2} \sum_{1 \leq i,j \leq N} \left| \left( \frac{d(x_i, x_j)}{D_X} - \frac{d(y_i, y_j)}{D_Y} \right) + \left( \frac{d(y_i, y_j)}{D_Y} - \frac{d(z_i, z_j)}{D_Z} \right) \right|.$$

Since $|a + b| \leq |a| + |b|$,

$$\mathsf{NSA}(X, Z) \leq \frac{1}{N^2} \sum_{1 \leq i,j \leq N} \left( \left| \frac{d(x_i, x_j)}{D_X} - \frac{d(y_i, y_j)}{D_Y} \right| + \left| \frac{d(y_i, y_j)}{D_Y} - \frac{d(z_i, z_j)}{D_Z} \right| \right).$$

Hence,

$$\mathsf{NSA}(X, Z) \leq \frac{1}{N^2} \sum_{1 \leq i,j \leq N} \left| \frac{d(x_i, x_j)}{D_X} - \frac{d(y_i, y_j)}{D_Y} \right| + \frac{1}{N^2} \sum_{1 \leq i,j \leq N} \left| \frac{d(y_i, y_j)}{D_Y} - \frac{d(z_i, z_j)}{D_Z} \right|,$$

or $\mathsf{NSA}(X, Z) \leq \mathsf{NSA}(X, Y) + \mathsf{NSA}(Y, Z)$. $\qquad \square$

From Lemma 1, Lemma 2, Lemma 3 and Lemma 4, we see that NSA is a psuedometric over the space of point clouds.

## B  PROOFS FOR NSA AS A SIMILARITY METRIC

### B.1  INVARIANCE TO ISOTROPIC SCALING

**Lemma 5** (Invariance to Isotropic scaling in the first coordinate). *Let $X$ and $Y$ be two point clouds of the same size. Let $c \in \mathbf{R}$ and $c \neq 0$, $X_c$ be the point cloud with each point in $X$ scaled by a factor of c, then $\mathsf{NSA}(X, Y) = \mathsf{NSA}(X_c, Y)$.*

*Proof.* Let $X = \{x_1, \ldots, x_N\}$ and $Y = \{y_1, \ldots, y_N\}$, then $X_c = \{cx_1, \ldots, cx_N\}$. Let $\max_{x \in X}(d(x, 0)) = D_X$, $\max_{y \in Y}(d(y, 0)) = D_Y$ and $\max_{x \in X_c}(d(x, 0)) = D_{X_c}$. Since, each point in $X_c$ is the $c$ times each point in $X$. Then, we can write $D_{X_c} = \max_{x \in X}(d(cx, 0))$. Since, for any two points, $x_1$ and $x_2$, $d(cx_1, cx_2) = |c|d(x_1, x_2)$, then $D_{X_c} = |c| \max_{x \in X}(d(x, 0))$. Hence, $D_{x_c} = |c|D_X$. From the definition of NSA,

$$\mathsf{NSA}(X_c, Y) = \frac{1}{N^2} \sum_{1 \leq i,j \leq N} \left| \left( \frac{d(cx_i, cx_j)}{D_{X_c}} - \frac{d(y_i, y_j)}{D_Y} \right) \right|.$$

Again, using $d(cx_i, cx_j) = |c|d(x_i, x_j)$ and $D_{X_c} = |c|D_X$,

$$\mathsf{NSA}(X_c, Y) = \frac{1}{N^2} \sum_{1 \leq i,j \leq N} \left| \left( \frac{|c|d(x_i, x_j)}{|c|D_X} - \frac{d(y_i, y_j)}{D_Y} \right) \right|.$$

Simplifying,

$$\mathsf{NSA}(X_c, Y) = \frac{1}{N^2} \sum_{1 \leq i,j \leq N} \left| \left( \frac{d(x_i, x_j)}{D_X} - \frac{d(y_i, y_j)}{D_Y} \right) \right|.$$

Hence, $\mathsf{NSA}(X_c, Y) = \mathsf{NSA}(X, Y)$. $\qquad \square$

By Lemma 2, we can see that this gives us invariance to isotropic scaling in the second coordinate as well.

**Lemma 6** (Invariance to Isotropic scaling in the second coordinate). *Let $X$ and $Y$ be two point clouds of the same size. Let $c \in \mathbf{R}$ and $c \neq 0$, $Y_c$ be the point cloud with each point in $Y$ scaled by a factor of c, then $\mathsf{NSA}(X, Y) = \mathsf{NSA}(X, Y_c)$.*

Combining Lemma 5 and *Lemma* 6, we get the required lemma.

**Lemma 7** (Invariance to Isotropic scaling). *Let $X$ and $Y$ be two point clouds of the same size. Let $c_1, c_2 \in \mathbf{R}$, $c_1 \neq 0$ and $c_2 \neq 0$, $X_{c_1}$ be the point cloud with each point in $X$ scaled by a factor of $c_1$ and $Y_{c_2}$ be the point cloud with each point in $Y$ scaled by a factor of $c_2$, then $\mathsf{NSA}(X, Y) = \mathsf{NSA}(X_{c_1}, Y_{c_2})$.*

### B.2 INVARIANCE TO ORTHOGONAL TRANSFORMATION

**Lemma 8** (Invariance to Orthogonal transformation in the first coordinate). *Let $X$ and $Y$ be two point clouds of the same size. Let $U$ be an orthogonal transformation on the point space of $X$, $X_U$ be the point cloud with each point in $X$ transformed using $U$, then $\mathsf{NSA}(X,Y) = \mathsf{NSA}(X_U,Y)$.*

*Proof.* Let $X = \{x_1,\ldots,x_N\}$ and $Y = \{y_1,\ldots,y_N\}$, then $X_U = \{Ux_1,\ldots,Ux_N\}$. Let $\max_{x\in X}(d(x,0)) = D_X$, $\max_{y\in Y}(d(y,0)) = D_Y$ and $\max_{x\in X_U}(d(x,0)) = D_{X_U}$. Since, each point in $X_U$ is the $U$ times each point in $X$. Then, we can write $D_{X_U} = \max_{x\in X}(d(Ux,0))$. Since, for any two points, $x_1$ and $x_2$, $d(Ux_1,x_2) = d(x_1,U^T x_2)$, and $U^T 0 = 0$, then $D_{X_c} = \max_{x\in X}(d(x,0))$. Hence, $D_{x_c} = D_X$. From the definition of NSA,

$$\mathsf{NSA}(X_U,Y) = \frac{1}{N^2}\sum_{1\le i,j\le N}\left|\left(\frac{d(Ux_i,Ux_j)}{D_{X_c}} - \frac{d(y_i,y_j)}{D_Y}\right)\right|.$$

Again, using $d(Ux_i,Ux_j) = d(x_i,U^T Ux_j)$, and $U^T U = I$,

$$\mathsf{NSA}(X_U,Y) = \frac{1}{N^2}\sum_{1\le i,j\le N}\left|\left(\frac{d(x_i,x_j)}{D_X} - \frac{d(y_i,y_j)}{D_Y}\right)\right|.$$

Hence, $\mathsf{NSA}(X_U,Y) = \mathsf{NSA}(X,Y)$. $\qquad\square$

By Lemma 2, we get invariance to orthogonal transformation in the second coordinate too and combining, we get the required lemma.

**Lemma 9** (Invariance to Orthogonal transformation). *Let $X$ and $Y$ be two point clouds of the same size. Let $U_1$ be an orthogonal transformation on the point space of $X$, $X_{U_1}$ be the point cloud with each point in $X$ transformed using $U_1$. Similarly, let $U_2$ be an orthogonal transformation on the point space of $Y$, $Y_{U_2}$ be the point cloud with each point in $Y$ transformed using $U_2$, then $\mathsf{NSA}(X,Y) = \mathsf{NSA}(X_{U_1},Y_{U_2})$.*

### B.3 NOT INVARIANT UNDER INVERTIBLE LINEAR TRANSFORMATION (ILT)

We can easily see this with a counter example. Let $X = \{(1,0,0),(0,1,0),(0,0,1)\}$. Let $Y = \{(1,1,0),(1,0,1),(0,1,1)\}$. Define $A$ to be an invertible linear map such that $A(1,0,0) = (2,0,0)$, $A(0,1,0) = (0,1,0)$ and $A(0,0,1) = (0,0,1)$. Let $X_A$ be the point cloud with each point in $X$ transformed using $A$. Then from some calculation, we can see that $\mathsf{NSA}(X,Y) = 0$ but $\mathsf{NSA}(X_A,Y) = \frac{1}{9}$. Hence, we can see that NSA is not invariant under Invertible Linear Transformations.

## C PROOFS FOR NSA AS A LOSS FUNCTION

### C.1 DIFFERENTIABILITY OF NSA

To derive the sub-gradient $\frac{\partial \mathsf{NSA}(X,Y)}{\partial x_i}$, we first define the following notation. Let $\mathbb{I} : \{T,F\} \to \{0,1\}$ be a function that takes as input some condition and outputs 0 or 1 such that $\mathbb{I}(T) = 1$ and $\mathbb{I}(F) = 0$. Let $D_X = \max_{x\in X}(d(x,0))$. Then it is easy to see that

$$\frac{\partial D_X}{\partial x_i} = \frac{x_i}{d(x_i,0)}\mathbb{I}\{\arg\max_{x\in X}(d(x,0)) = i\}.$$

Also, notice that $\frac{\partial d(x_i,x_j)}{x_i} = \frac{x_i - x_j}{d(x_i,x_j)}$, and that for $j \ne i$ and $k \ne i$, $\frac{\partial d(x_j,x_k)}{x_i} = 0$.

Let $m_X^{i,j} = \frac{d(x_i,x_j)}{D_X}$ and $m_Y^{i,j} = \frac{d(y_i,y_j)}{D_Y}$. Hence, combining, we get

$$\frac{\partial m_X^{j,k}}{\partial x_i} = \frac{\frac{\partial d(x_j,x_k)}{\partial x_i}}{D_X} - \frac{d(x_j,x_k)\frac{\partial D_X}{\partial x_i}}{(D_X)^2}, \text{ where these partial differentials are derived above.}$$

Lastly, notice that $\frac{\partial|m_X^{j,k}-m_Y^{j,k}|}{\partial m_X^{j,k}} = (-1)^{\mathbb{I}(m_Y^{j,k}>m_X^{j,k})}$.

Combining all the above, we get

$$\frac{\partial \mathsf{NSA}(X,Y)}{\partial x_i} = \frac{1}{N^2} \sum_{1 \leq j,k \leq N} \frac{\partial \left| m_X^{j,k} - m_Y^{j,k} \right|}{\partial m_X^{j,k}} \frac{\partial m_X^{j,k}}{\partial x_i} = \frac{1}{N^2} \sum_{1 \leq j,k \leq N} (-1)^{\mathbb{I}(m_Y^{j,k} > m_X^{j,k})} \frac{\partial m_X^{j,k}}{\partial x_i}.$$

Sub-gradients $\frac{\partial \mathsf{NSA}(X,Y)}{\partial y_i}$ can be similarly derived.

## C.2 Continuity

We prove the continuity of NSA by showing that NSA is a composition of continuous functions (Carothers, 2000, Theorem 5.10). Let

$$f_{i,j}(X,Y) = \left| \frac{d(x_i, x_j)}{\max_{x \in X}(d(x,0))} - \frac{d(y_i, y_j)}{\max_{y \in Y}(d(y,0))} \right|.$$

It's easy to see that $\qquad \mathsf{NSA}(X,Y) = \frac{1}{N^2} \sum_{1 \leq i,j \leq N} f_{i,j}(X,Y).$

Since a sum of continuous functions is continuous, all we need to show is that $f_{i,j}(X,Y)$ is continuous for all $1 \leq i,j \leq N$. This is easy to see because $f_{i,j}(X,Y)$ can be seen as a composition of $d(\cdot,\cdot)$ and $\max(\cdot)$, along with the algebraic operations of subtraction, division and taking the absolute value. All the above operations are continuous everywhere[2], we get that their composition is also continuous everywhere. Hence, $\mathsf{NSA}(\cdot,\cdot)$ is continuous.

## D Convergence of subset NSA

In the following section, we show that NSA corresponds well to minibatching and hence can be used as a loss term. In particular, Lemma lemma 10 proves theoretically that in expectation, NSA over a minibatch is equal to NSA of the whole dataset. Figure 6 shows that NSA over a large number of trials approximates the NSA of the entire dataset while RTD fails to do so. The experiment is run on the output embeddings of a GCN trained on node classification for the Cora Dataset. We compare the convergence with a batch size of 200 and 500.

**Lemma 10** (Subset NSA convergence). *Let $X, Y$ be two point clouds of the same size $N$ such that $X = \{x_1, \ldots, x_N\}$ and $Y = \{y_1, \ldots, y_N\}$. Let $\tilde{X}$ be some randomly sampled $s$ sized subsets of $X$ (for some $s \leq N$). Define $\tilde{Y} \subset Y$ as $y_i \in \tilde{Y}$ iff $x_i \in \tilde{X}$. Then the following holds true*

$$\mathop{\mathbb{E}}_{\tilde{X}, \tilde{Y}} \left[ \mathsf{NSA}(\tilde{X}, \tilde{Y}) \right] = \mathsf{NSA}(X, Y).$$

*Proof.* Let $\max_{x \in X}(d(x,0)) = D_X$ and $\max_{y \in Y}(d(y,0)) = D_Y$. Then

$$\mathsf{NSA}(X,Y) = \frac{1}{N^2} \sum_{1 \leq i,j \leq N} \left| \frac{d(x_i, x_j)}{D_X} - \frac{d(y_i, y_j)}{D_Y} \right|.$$

Define $\mathbb{I}_{\tilde{X}}(x_i)$ as

$$\mathbb{I}_{\tilde{X}}(x_i) = \begin{cases} 1 \text{ if } x_i \in \tilde{X}, \\ 0 \text{ otherwise.} \end{cases}$$

Then we can see that

$$\mathsf{NSA}(\tilde{X}, \tilde{Y}) = \frac{1}{s^2} \sum_{1 \leq i,j \leq N} \left| \frac{d(x_i, x_j)}{D_X} - \frac{d(y_i, y_j)}{D_Y} \right| \mathbb{I}_{\tilde{X}}(x_i) \mathbb{I}_{\tilde{X}}(x_j).$$

---

[2]Note that division is not continuous when the denominator goes to zero. Since, the denominators are $\max_{x \in X}(d(x,0))$ and $\max_{y \in Y}(d(y,0))$, as long as neither of the representations is all zero, the denominator does not go to zero.

Hence, taking expectation over $\tilde{X}, \tilde{Y}$, we get

$$\underset{\tilde{X},\tilde{Y}}{\mathbb{E}} \left[ \mathsf{NSA}(\tilde{X}, \tilde{Y}) \right] = \frac{1}{s^2} \underset{\tilde{X},\tilde{Y}}{\mathbb{E}} \left[ \sum_{1 \leq i,j \leq N} \left| \frac{d(x_i, x_j)}{D_X} - \frac{d(y_i, y_j)}{D_Y} \right| \mathbb{I}_{\tilde{X}}(x_i) \, \mathbb{I}_{\tilde{X}}(x_j) \right].$$

By linearity of expectation, we get

$$\underset{\tilde{X},\tilde{Y}}{\mathbb{E}} \left[ \mathsf{NSA}(\tilde{X}, \tilde{Y}) \right] = \frac{1}{s^2} \sum_{1 \leq i,j \leq N} \left| \frac{d(x_i, x_j)}{D_X} - \frac{d(y_i, y_j)}{D_Y} \right| \underset{\tilde{X},\tilde{Y}}{\mathbb{E}} \left[ \mathbb{I}_{\tilde{X}}(x_i) \, \mathbb{I}_{\tilde{X}}(x_j) \right].$$

Assuming $s >> 1$, each $x_i$ is independently in $\tilde{X}$ with probability $s/N$, hence, we get

$$\underset{\tilde{X},\tilde{Y}}{\mathbb{E}} \left[ \mathsf{NSA}(\tilde{X}, \tilde{Y}) \right] = \frac{1}{s^2} \sum_{1 \leq i,j \leq N} \left| \frac{d(x_i, x_j)}{D_X} - \frac{d(y_i, y_j)}{D_Y} \right| \frac{s}{N} \frac{s}{N}.$$

Hence, we have

$$\underset{\tilde{X},\tilde{Y}}{\mathbb{E}} \left[ \mathsf{NSA}(\tilde{X}, \tilde{Y}) \right] = \mathsf{NSA}(X, Y).$$

$\square$

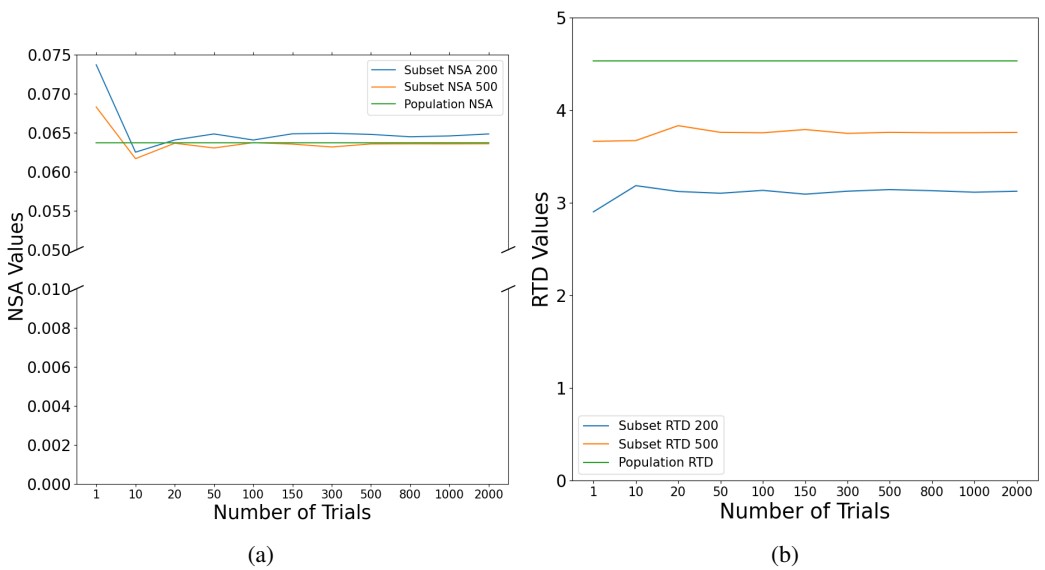

(a)                    (b)

Figure 6: Expectation of subset metrics over a large number of trials. (a) Mean subset NSA variation as the number of trials are increased. (b) Mean subset RTD variation as the number of trials are increased.

## E    ARE GNNS TASK SPECIFIC LEARNERS?

We use NSA to investigate whether representations produced by GNNs are specific to the downstream task. For this, we explored the similarity of representations of different GNN architectures when trained on the same task versus their similarity when trained on different downstream tasks.

### E.1    CROSS ARCHITECTURE TESTS ON THE SAME DOWNSTREAM TASK

We tested layerwise representational similarity on the task of node classification (on the Amazon Computers Dataset) across different GNN architectures: GCN, ClusterGCN (CGCN), GraphSAGE, and GAT. Just like in the sanity tests, models that are similar in architecture or training paradigms have a higher layerwise similarity: this implies that the highest similarity is between GCN and ClusterGCN that differ only in their training paradigms. We also observe (a less pronounced) linear relationship between the other pairs of architectures. These results are shown in Figure 7.

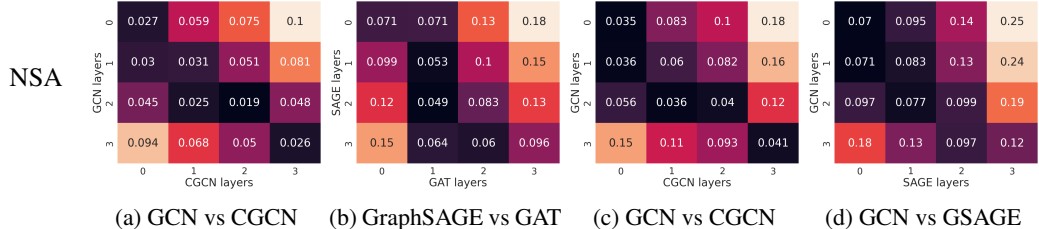

|  | (a) GCN vs CGCN | (b) GraphSAGE vs GAT | (c) GCN vs CGCN | (d) GCN vs GSAGE |

Figure 7: Cross Architecture Tests using NSA on the Amazon Computers Dataset. (a) Layerwise NSA values between GCN and ClusterGCN on Node Classification (b) Layerwise NSA values between GraphSAGE and GAT on Node Classification (c) Layerwise NSA values between GCN and ClusterGCN on Link Prediction (d) Layerwise NSA values between GCN and GSAGE on Link Prediction. Similar architectures showcase a layerwise pattern when trained on the same task.

## E.2 ARCHITECTURE TESTS ON DIFFERENT DOWNSTREAM TASKS

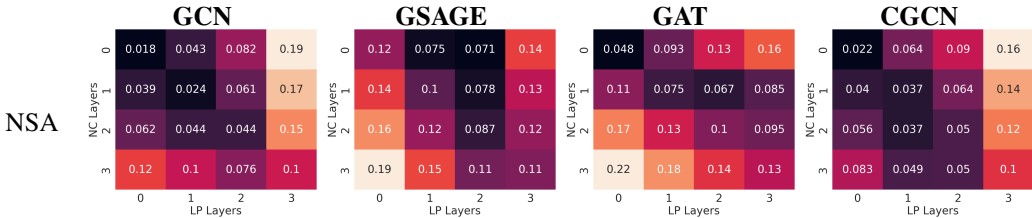

Figure 8: Effect of Downstream Task on Representations Achieved by the Same Architecture (Amazon Computers Dataset). The layerwise dissimilarity of four GNN architectures is compared on two different downstream tasks: Node Classification and Link Prediction. There is no observable correlation across layers suggesting that the GNNs generate different representation spaces for the same dataset on different downstream tasks.

We conducted experiments with different downstream tasks to assess layerwise similarity between two models, both of which shared structural identity except for disparities in their final layers. Specifically, we employed node classification and link prediction as the two downstream tasks for our investigations. We present our findings in Figure 8. They reveal that there is little relationship across layers for any of the four architectures.

The results presented in this section indicate that that different GNN architectures have similar representation spaces when trained on the same downstream task and conversely, similar architectures have different representation spaces when trained on different downstream tasks. Extending this idea further, it should be possible to train Graph Neural Networks to conform to a task specific representation template. This structural template will be agnostic of GNN architectures and provide a high degree of functional similarity. If we train GNNs to minimize discrepancy loss with such a task specific template, we could train more directly and without adding downstream layers of tasks.

## F SANITY TESTS FOR LINK PREDICTION

We show the results for link prediction across GNN architectures for all three similarity metrics. We observe that although all 3 metrics pass the sanity test, NSA shows the best gradient in similarity across adjacent layers of some models like GAT and CGCN. The heatmaps for the sanity tests for Link Prediction are given in Figure 9

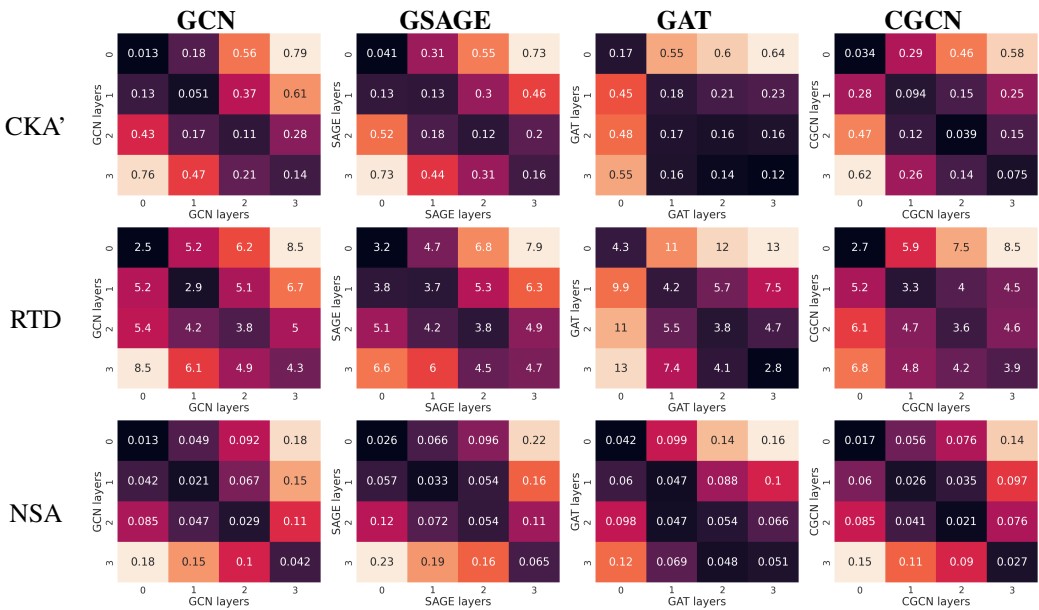

Figure 9: Sanity Tests for Link Prediction

# G  CROSS ARCHITECTURE TESTS ON AMAZON DATASET

## G.1  CROSS ARCHITECTURE TESTS WITH NSA

The cross architecture test results for NSA on link prediction and node classification for the architectures not shown in Figure 7 are given in Figure 10 and Figure 11. These are architectures with low degree of similarity hence the we do not observe a strong linear relationship across layers with NSA.

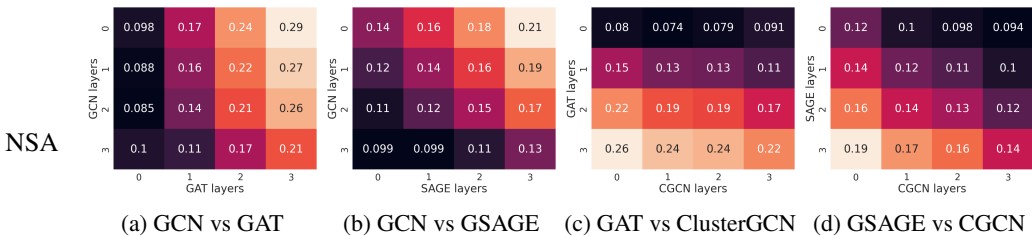

(a) GCN vs GAT   (b) GCN vs GSAGE   (c) GAT vs ClusterGCN   (d) GSAGE vs CGCN

Figure 10: Cross Architecture Tests using Normalized Space Alignment for Node Classification for the remaining architectures

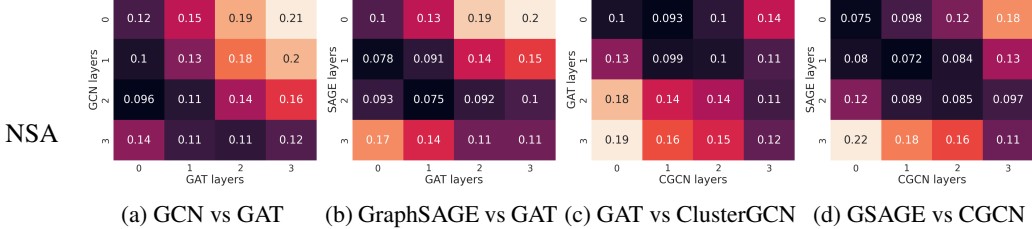

(a) GCN vs GAT   (b) GraphSAGE vs GAT  (c) GAT vs ClusterGCN   (d) GSAGE vs CGCN

Figure 11: Cross Architecture Tests using Normalized Space Alignment for Link Prediction for the remaining four architectures

## G.2 Cross Architecture Tests with RTD and CKA' on Node Classification

The cross architecture test results for CKA' and RTD on Node Classification are shown in Figure 12 and Figure 13

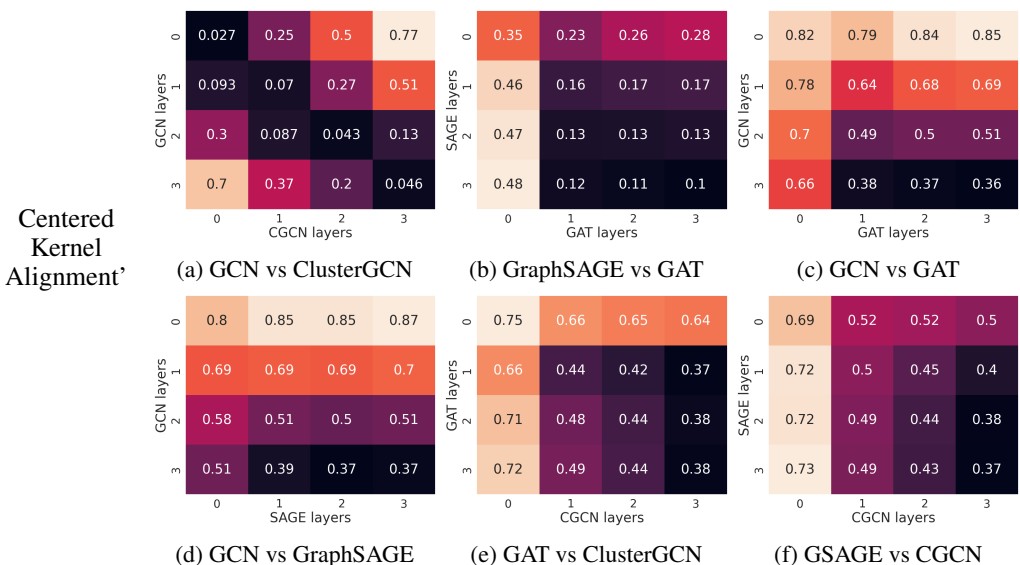

Centered Kernel Alignment'

(a) GCN vs ClusterGCN     (b) GraphSAGE vs GAT     (c) GCN vs GAT

(d) GCN vs GraphSAGE     (e) GAT vs ClusterGCN     (f) GSAGE vs CGCN

Figure 12: Cross Architecture Tests using Centered Kernel Alignment' for Node Classification

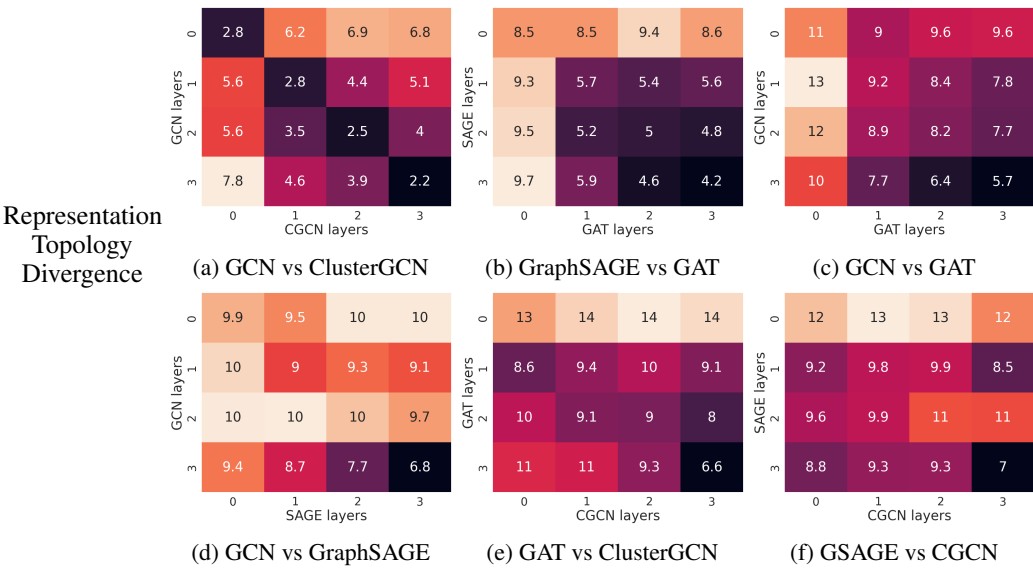

Representation Topology Divergence

(a) GCN vs ClusterGCN     (b) GraphSAGE vs GAT     (c) GCN vs GAT

(d) GCN vs GraphSAGE     (e) GAT vs ClusterGCN     (f) GSAGE vs CGCN

Figure 13: Cross Architecture Tests using Representation Topology Divergence for Node Classification

## G.3 Cross Architecture Tests with RTD and CKA' on Link Prediction

The cross architecture test results for CKA' and RTD on Link Prediction are shown in Figure 14 and Figure 15. We observe that CKA' and RTD do not show a layerwise pattern as well as NSA does. We can observe that RTD manages to capture the low dissimilarity between corresponding layers of

GCN and ClusterGCN and for some other models but CKA' fails to capture the layerwise similarity for any architecture combination.

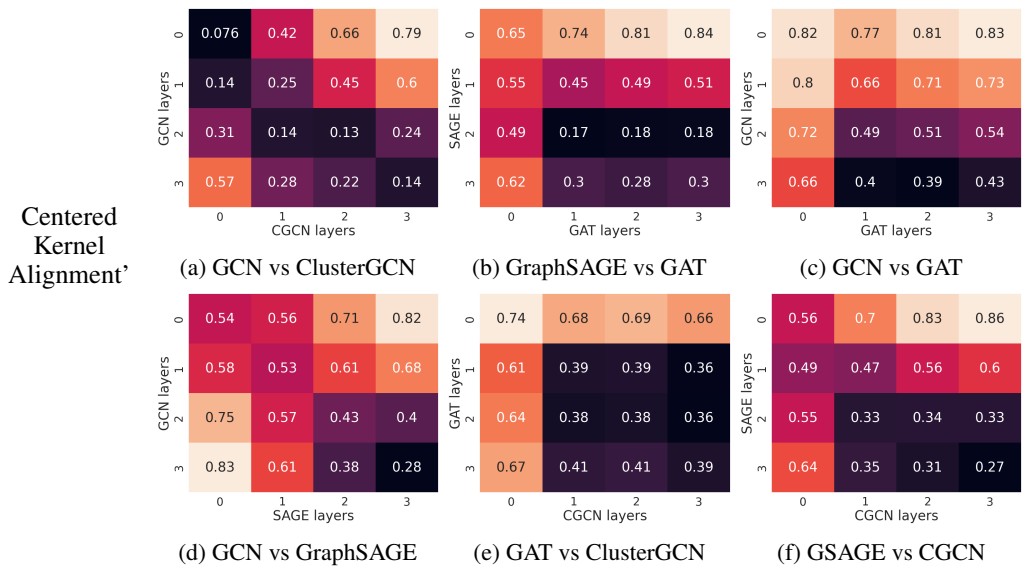

Figure 14: Cross Architecture Tests using Centered Kernel Alignment' for Link Prediction

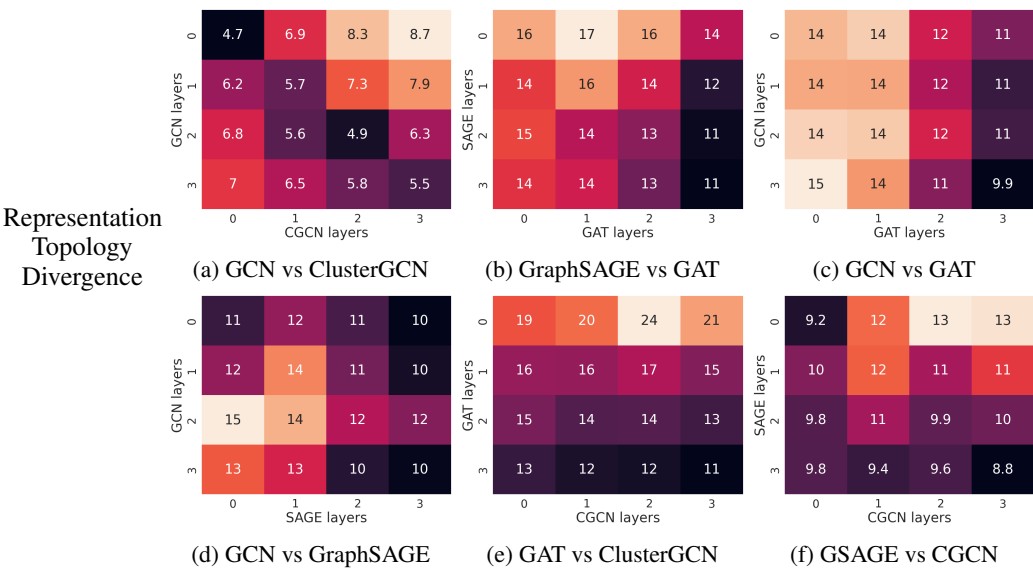

Figure 15: Cross Architecture Tests using Representation Topology Divergence for Link Prediction

## H    CROSS DOWNSTREAM TASK FOR CKA' AND RTD

We show the performance of CKA' and RTD when we test how GNN architectures compare across different downstream tasks. Similar to NSA, CKA' and RTD fail to show any noticeable pattern across layers. The heatmaps for cross downstream tasks with CKA' and RTD are given in Figure 16

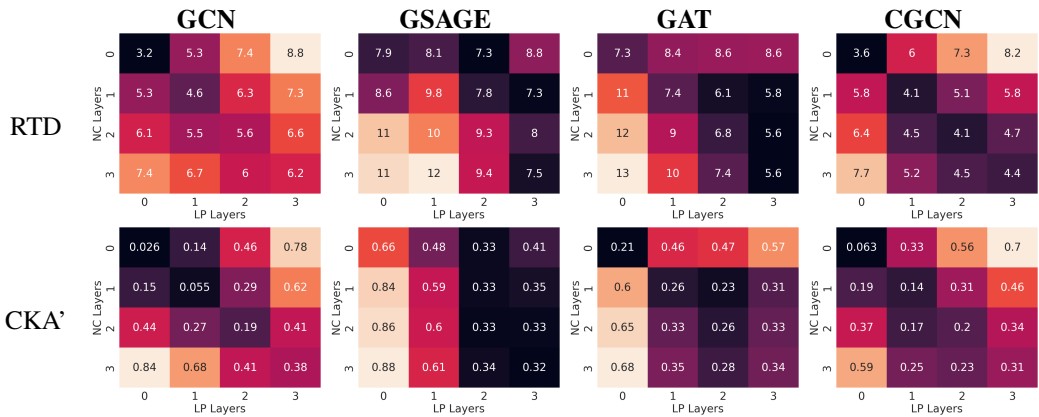

Figure 16: Cross Downstream Task Tests CKA′ and RTD

# I ADDITIONAL CONVERGENCE TESTS ON AMAZON DATASET

We show the convergence tests with GCN and GraphSAGE on Node Classification and Link Prediction in Figure 17. Link Prediction models were trained for 200 epochs and show similar patterns to the ones observed with node classification. All convergence tests are performed with NSA only.

# J RUNNING TIME AND RECONSTRUCTION LOSS FOR AUTOENCODER

We show the results in Table 3

| Dataset | Method | Metric | | |
| | | Time per epoch | Train MSE | Test MSE |
|---|---|---|---|---|
| MNIST | AE | 2.344 | 7.64e-3 | 8.62e-3 |
| | TopoAE | 39.168 | 6.89e-3 | 8.16e-3 |
| | RTD-AE | 51.608 | 9.36e-3 | 1.07e-2 |
| | NSA-AE | 5.816 | 8.75e-3 | 1.00e-2 |
| F-MNIST | AE | 6.436 | 8.99e-03 | 9.79e-03 |
| | TopoAE | 37.26 | 8.94-03 | 9.83e-03 |
| | RTD-AE | 59.94 | 1.12e-02 | 1.24e-02 |
| | NSA-AE | 5.764 | 9.49e-03 | 1.04e-02 |
| CIFAR-10 | AE | 5.16 | 1.56e-02 | 1.65e-02 |
| | TopoAE | 58.664 | 1.54e-02 | 1.68e-02 |
| | RTD-AE | 56.172 | 1.60e-02 | 1.91e-02 |
| | NSA-AE | 6.996 | 1.58e-02 | 1.71e-02 |
| COIL-20 | AE | 2.012 | 1.67e-02 | - |
| | TopoAE | 4.876 | 1.09e-02 | - |
| | RTD-AE | 16.404 | 1.90e-02 | - |
| | NSA-AE | 8.716 | 1.80e-02 | - |

Table 3: Reconstruction Loss and Time Per Epoch for different Autoencoder architectures. All architectures were trained for 250 epochs with the auxiliary loss (TopoLoss, RTD, NSA) kicking in after 60 epochs.

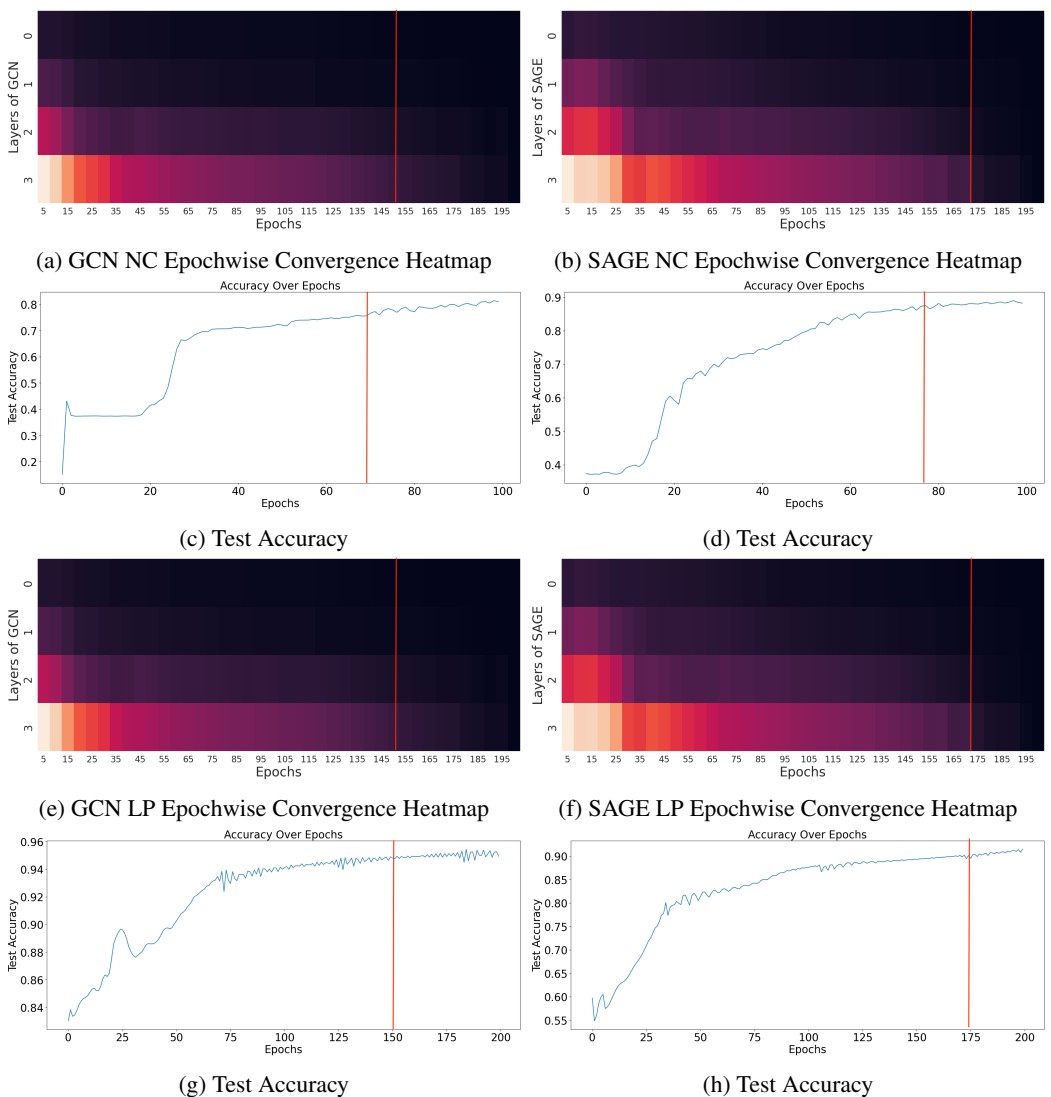

Figure 17: Convergence Tests on Graph Convolution Network and GraphSAGE

## K    AUTOENCODER HYPERPARAMETERS

The hyperparameter setup for all the autoencoder architectures is detailed in Table 4. All autoencoder architectures are trained using pytorch-lightning and the input data is normalized using a MinMaxScaler.

| Dataset Name | Batch Size | LR | Hidden Dim | Layers | Epochs | Metric Start Epoch |
|---|---|---|---|---|---|---|
| MNIST | 256 | $10^{-4}$ | 512 | 3 | 250 | 60 |
| F-MNIST | 256 | $10^{-4}$ | 512 | 3 | 250 | 60 |
| CIFAR-10 | 256 | $10^{-4}$ | 512 | 3 | 250 | 60 |
| COIL-20 | 256 | $10^{-4}$ | 512 | 3 | 250 | 60 |

Table 4: Autoencoder Hyperparameters. All four architectures used the same hyperparameters. To ensure similarity of testing conditions we replicate the hyperparameter setup from Trofimov et al. (2023)

## L VISUALIZING THE LATENT EMBEDDINGS FROM AUTOENCODERS

We use t-SNE(van der Maaten & Hinton, 2008) to reduce the latent embeddings obtained from all 4 datasets used in Table 1 to reduce their dimension from 16 to 3. The results are presented in Figure 18. We observe that t-SNE generates similar clusters across all three architectures; the basic Autoencoder, RTD-Autoencoder and NSA-Autoencoder. The T-SNE algorithm used default parameters to reduce the latent embeddings to 3D.

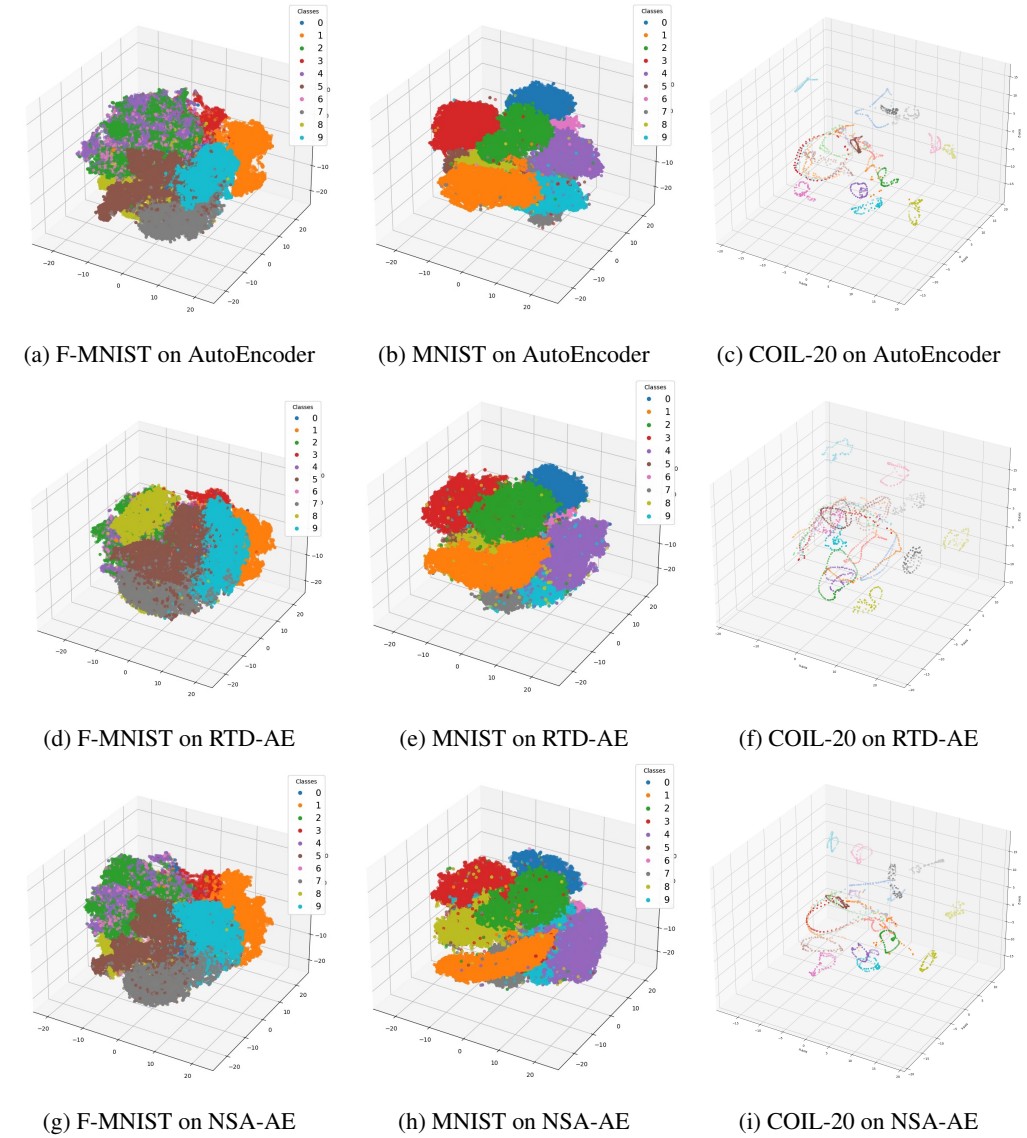

Figure 18: Visualizing the latent representations of the autoencoders. All latent embeddings are in 16 dimensions before being reduced to 3 by tSNE

## M DOWNSTREAM TASK ANALYSIS WITH THE STS DATASET

We assess the performance of embeddings derived from NSA-AE in the context of the Semantic Text Similarity (STS) Task. We employ Google's Word2Vec word vectors as the initial dataset, processing these 300-dimensional vectors through both a standard autoencoder and the NSA-Autoencoder to achieve reduced dimensions of 128 and 64. The efficacy of these dimensionality-reduced embeddings is then evaluated using the STS Multi EN dataset (Cer et al., 2017). This evaluation aims to

determine if the word vectors, once dimensionally reduced, maintain their alignment, as evidenced by a strong positive correlation in similarity scores on the STS dataset. The comparative results, as detailed in Table 5, demonstrate that the NSA-Autoencoder outperforms the traditional autoencoder in preserving semantic accuracy at both 128 and 64 dimensions.

| Latent Dimension | Basic AutoEncoder | NSA AutoEncoder |
|---|---|---|
| 128 | -0.518 | 0.7632 |
| 64 | 0.567 | 0.7619 |

Table 5: Pearson Correlation between the similarities output from the original embeddings and the similarities output from the reduced embeddings. NSA-AE shows a much higher correlation with the similarity outputs obtained by evaluating the original word2vec embeddings on the STS multi EN dataset

## N  GNN HYPERPARAMETERS

The hyperparameter setup for all the GNN architectures is detailed in Table 6

| Architecture | Layers | Hidden Dim | LR | Epochs | Additional Info |
|---|---|---|---|---|---|
| GCN | 4 | 128 | 0.001 | 200 | - |
| GraphSAGE | 4 | 128 | 0.001 | 200 | Mean Aggregation |
| GAT | 4 | 128 | 0.001 | 200 | 8 heads with Hid Dim 8 each |
| CGCN | 4 | 128 | 0.001 | 200 | 8 subgraphs |

Table 6: GNN Architecture Information

## O  GNN METRICS

The test accuracy for node classification and the ROC AUC value for link prediction is detailed in Table 7

| Architecture | Accuracy (NC) | ROC AUC (LP) |
|---|---|---|
| GCN | 0.8257 | 0.8638 |
| GraphSAGE | 0.8800 | 0.8068 |
| GAT | 0.8273 | 0.7997 |
| CGCN | 0.8200 | 0.8716 |

Table 7: GNN Metric Data on Amazon Computer Dataset. Test Accuracy is for Node Classification and ROC AUC Score is for Link Prediction

## P  DATASETS

We report the statistics of the datasets used for the empirical analysis of GNNs in Table 8. We report the exact statistics of the autoencoder datasets in Table 9.

### P.1  GRAPH DATASETS

The Amazon computers dataset is a subset of the Amazon co-purchase graph (Shchur et al., 2019). The nodes represent products on amazon and the edges indicate that two products are frequently

bought together. The node features are the product reviews encoded in bag-of-words format. The class labels represent the product category. We also use the Flickr, Cora, Citeseer and Pubmed dataset in additional experiments.

| Dataset | Amazon Computer | Flickr | Cora | Citeseer | Pubmed |
|---|---|---|---|---|---|
| Number of Nodes | 13752 | 89250 | 2708 | 3327 | 19717 |
| Number of Edges | 491722 | 899756 | 10556 | 9104 | 88648 |
| Average Degree | 35.76 | 10.08 | 3.90 | 2.74 | 4.50 |
| Node Features | 767 | 500 | 1433 | 3703 | 500 |
| Labels | 10 | 7 | 7 | 6 | 3 |

Table 8: Dataset Statistics for Graph

## P.2 AUTOENCODER DATASETS

Four diverse real-world datasets were utilized for our experiments: MNIST (LeCun et al., 2010), Fashion-MNIST (F-MNIST) (Xiao et al., 2017), COIL-20 (Nene et al., 1996), and CIFAR-10 (Krizhevsky & Hinton, 2009), to comprehensively evaluate the performance of our autoencoder model. MNIST, comprising 28x28 grayscale images of handwritten digits, serves as a foundational benchmark for image classification and feature extraction tasks. Fashion-MNIST extends this by offering a similar format but with 10 classes of clothing items, making it an ideal choice for fashion-related image analysis. COIL-20 presents a unique challenge, with 20 object categories, where each category consists of 72 128x128 color images captured from varying viewpoints, offering a more complex 3D object recognition scenario. Lastly, CIFAR-10 introduces color and additional complexity, featuring 60,000 32x32 color images across 10 object classes, catering to real-world image analysis and deep learning challenges. Our experimentation across these datasets provides valuable insights into the versatility and effectiveness of our autoencoder approach for diverse tasks.

| Dataset | Classes | Train Size | Test Size | Image Size | Data Type |
|---|---|---|---|---|---|
| MNIST | 10 | 60,000 | 10,000 | 28x28 (784) | Grayscale |
| Fashion-MNIST (F-MNIST) | 10 | 60,000 | 10,000 | 28x28 (784) | Grayscale |
| COIL-20 | 20 | 1,440 | - | 128x128 (16384) | Color |
| CIFAR-10 | 10 | 60,000 | 10,000 | 32x32*3 (3072) | Color |

Table 9: Dataset Statistics for AE

## Q    SANITY TESTS FOR NODE CLASSIFICATION ON AMAZON DATASET

We show the results for node classification across GNN architectures for CKA'.

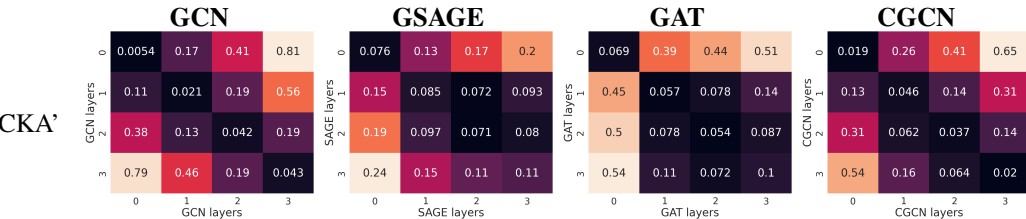

Figure 19: Sanity Tests for Node Classification for CKA′ on Amazon Computers Dataset

# R    EMPIRICAL ANALYSIS ON ADDITIONAL DATASETS

We utilize the Amazon Computers Dataset as our primary dataset throughout the empirical analyses conducted in the main text. In this section we evaluate the performance of NSA on other popular datasets. We evaluate with Sanity Tests, Cross Architecture Tests and Downstream Task tests. We utilize the Cora (McCallum et al., 2000), Citeseer (Giles et al., 1998), Pubmed (Sen et al., 2008) and Flickr (Zeng et al., 2020) Dataset for our experiments.

## R.1    SANITY TESTS

The sanity test compares the intermediate representations obtained from each layer of the Graph Neural Network against all the other layers of another GNN with the exact same architecture but trained with a different initialization. we expect that, for each layer's intermediate representation, the most similar intermediate representation in the other model should be the corresponding layer that matches in structure. We only demonstrate the performance of NSA for both Node Classification and Link Prediction.

### R.1.1    NODE CLASSIFICATION

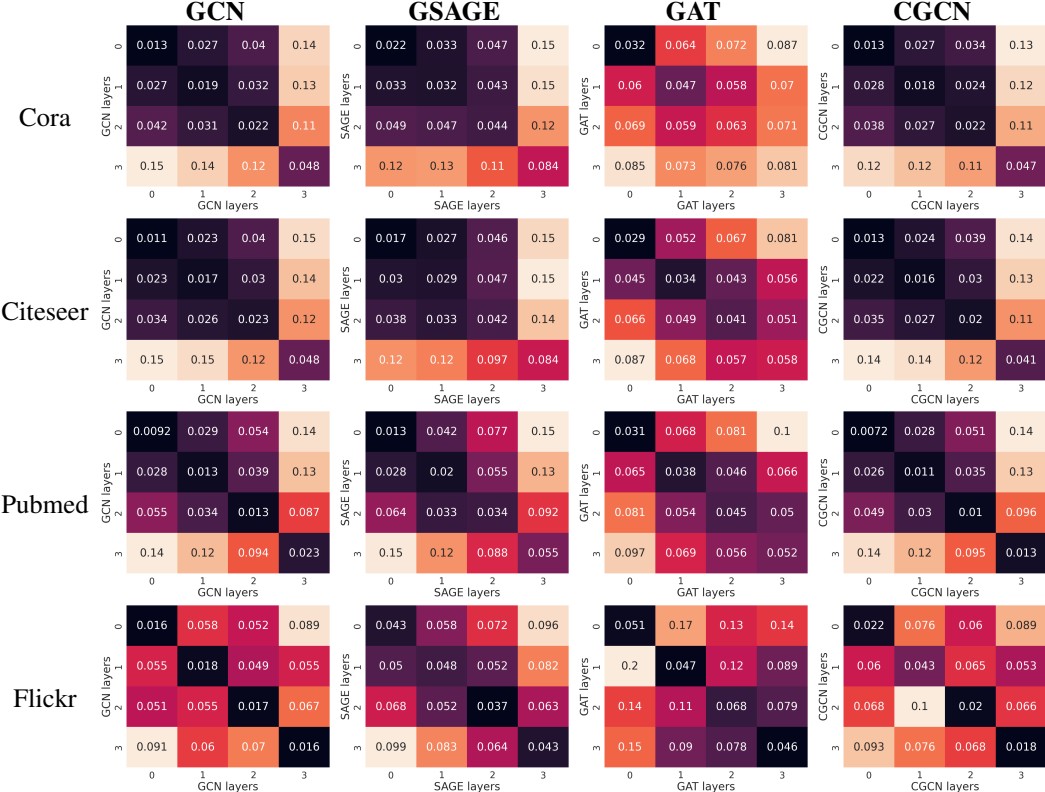

Figure 20: Sanity Tests for Node Classification on all 4 datasets. The heatmaps show layer-wise dissimilarity values for two different initialization of the same dataset on four different GNN architectures.

### R.1.2 LINK PREDICTION

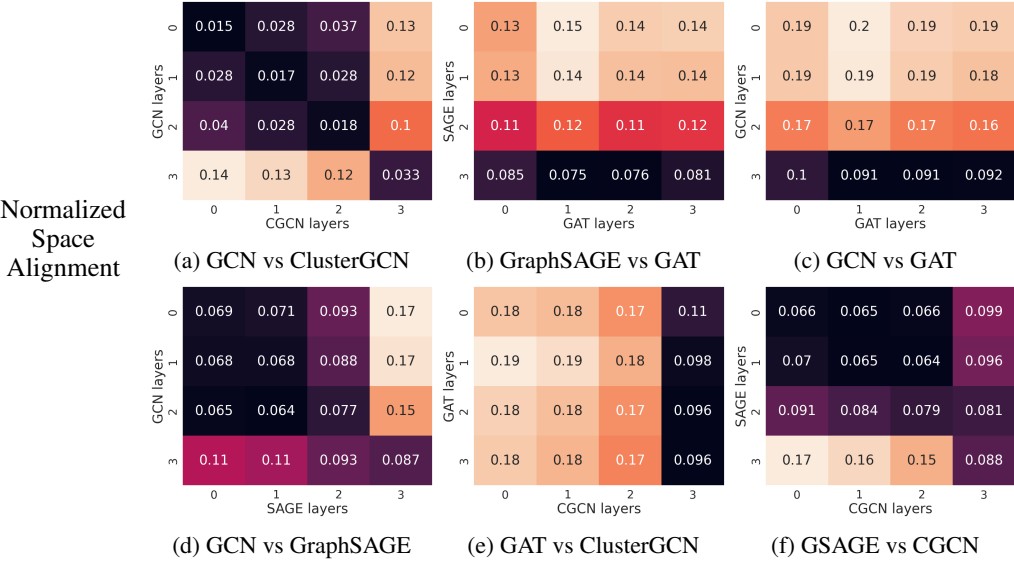

Figure 21: Sanity Tests for Link Prediction on 3 datasets. The heatmaps show layer-wise dissimilarity values for two different initialization of the same dataset on four different GNN architectures.

## R.2 CROSS ARCHITECTURE TESTS

In this section we evaluate the similarity in representations across architectures for the three Planetoid datasets; Cora, Citeseer and Pubmed on Node Classification and Link Prediction.

### R.2.1 NODE CLASSIFICATION

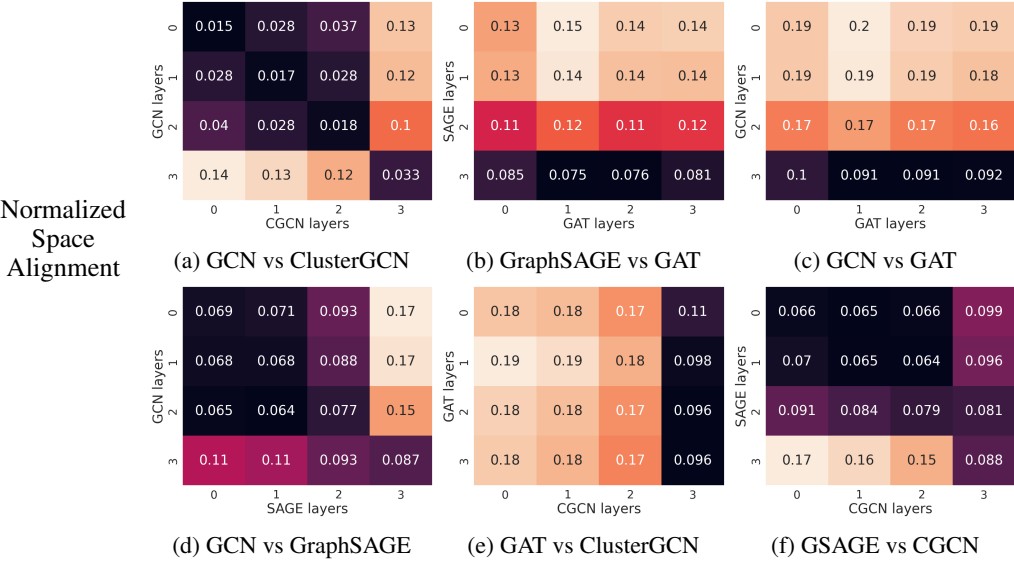

Figure 22: Cross Architecture Tests using NSA for Node Classification on the Cora Dataset

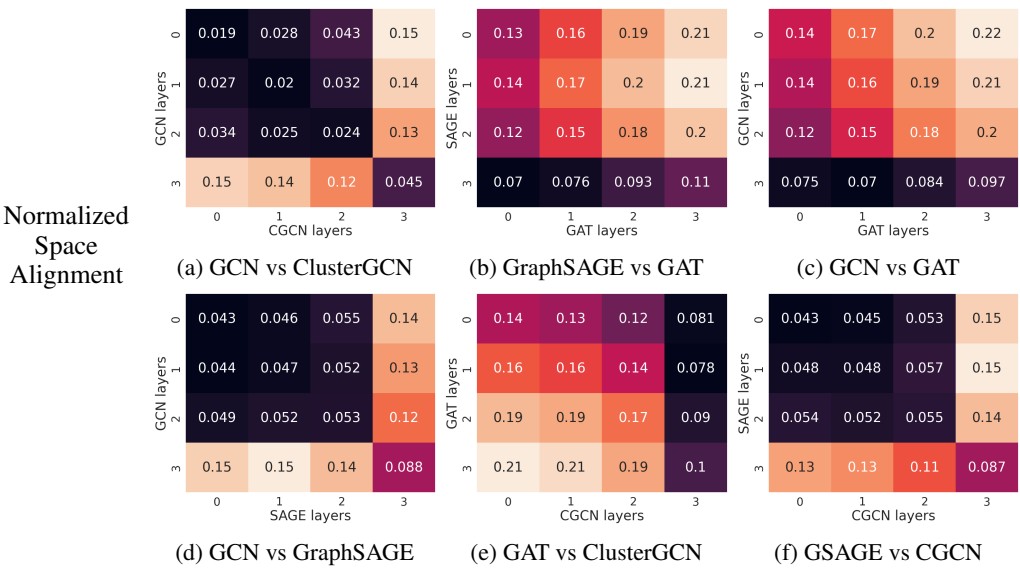

Figure 23: Cross Architecture Tests using NSA for Node Classification on the Citeseer Dataset

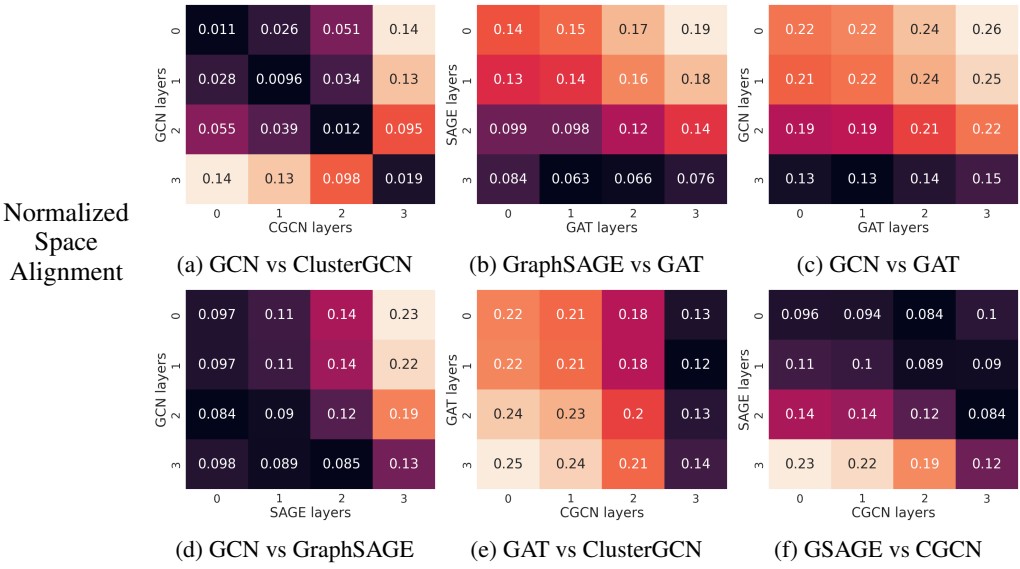

Figure 24: Cross Architecture Tests using NSA for Node Classification on the Pubmed Dataset

### R.2.2 LINK PREDICTION

Normalized
Space
Alignment

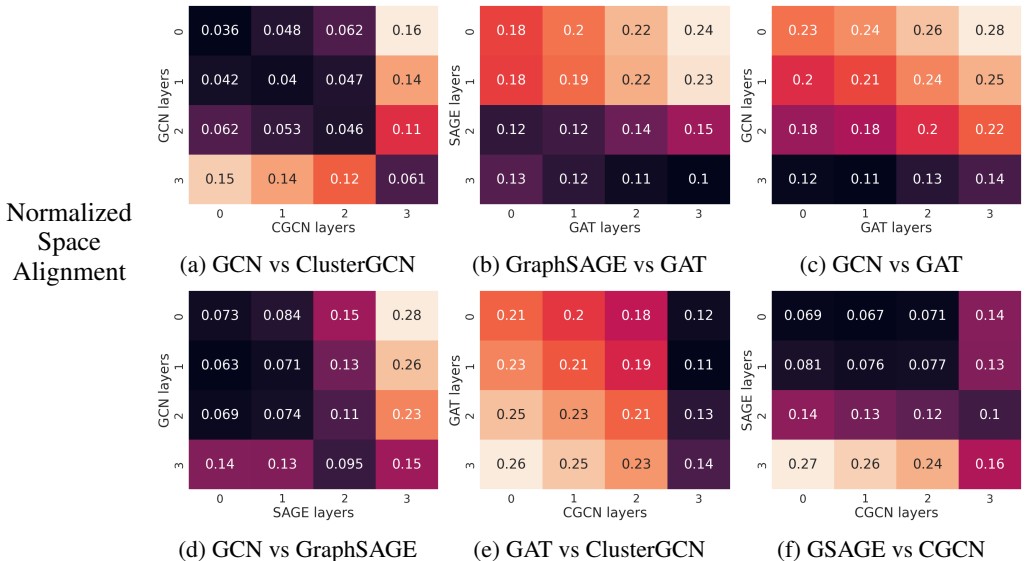

(a) GCN vs ClusterGCN  (b) GraphSAGE vs GAT  (c) GCN vs GAT

(d) GCN vs GraphSAGE  (e) GAT vs ClusterGCN  (f) GSAGE vs CGCN

Figure 25: Cross Architecture Tests using NSA for Link Prediction on the Cora Dataset

Normalized
Space
Alignment

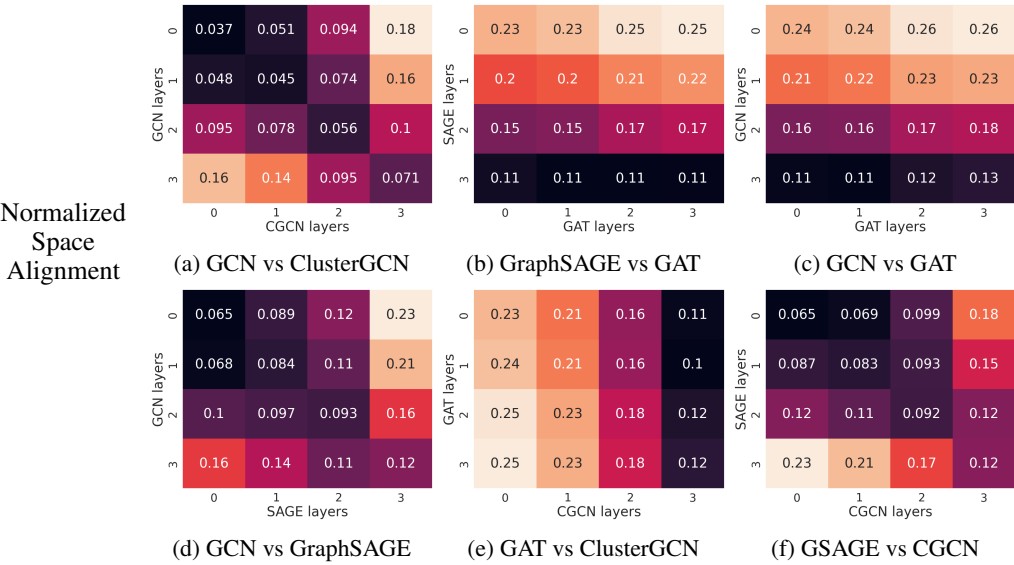

(a) GCN vs ClusterGCN  (b) GraphSAGE vs GAT  (c) GCN vs GAT

(d) GCN vs GraphSAGE  (e) GAT vs ClusterGCN  (f) GSAGE vs CGCN

Figure 26: Cross Architecture Tests using NSA for Link Prediction on the Citeseer Dataset

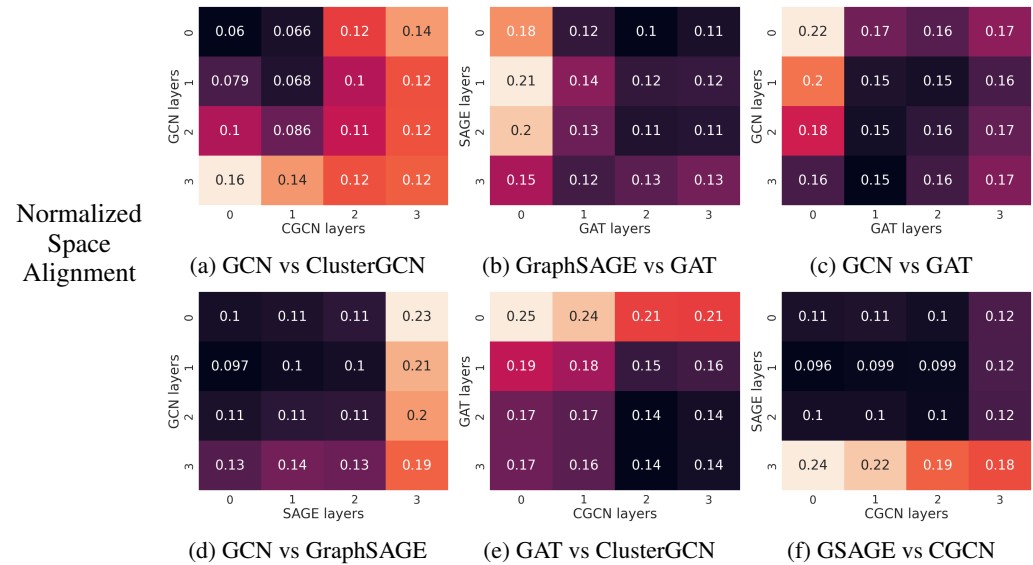

Figure 27: Cross Architecture Tests using NSA for Link Prediction on the Pubmed Dataset

## R.3 CROSS DOWNSTREAM TASK TESTS

Similar to our experiments with the Amazon dataset, we compare the similarity in the intermediate representations between those trained on Node Classification and Link Prediction on the same architecture. We use the Cora, Citeseer and Pubmed dataset for these experiments and evaluate the similarity with NSA.

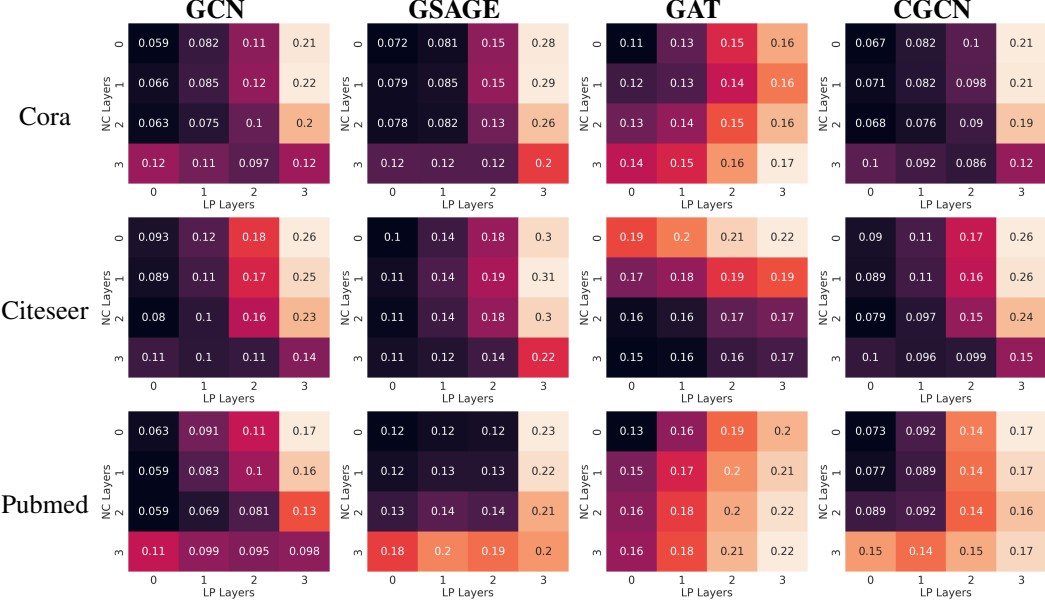

Figure 28: Cross Downstream Task Tests with NSA on the Cora, Citeseer and Pubmed Datasets

