# OpenReview forum: "Normalized Space Alignment: A Versatile Metric for Representation Space Discrepancy Minimization"
_ICLR.cc/2024/Conference — Submitted to ICLR 2024_

### Official Review · Reviewer_odB3 · 2023-10-18

**Soundness:** 3 good
**Presentation:** 2 fair
**Contribution:** 2 fair
**Rating:** 6
**Confidence:** 4

**Summary:**

This paper introduces a manifold analysis technique for quantifying the discrepancy between two representation spaces. The method is called Normalized Space Alignment (NSA).  NSA provides a robust means of comparing representations across different layers and models, a pseudometric that is both continuous and differentiable and an effective loss function for autoencoders. Empirical results show NSA consistently outperforms or matches previous techniques with high computational efficiency.

**Strengths:**

1. This paper is well-organized and easy to follow.
2. The proposed technique is sound. Building NSA upon Representational Similarity Matrix-Based Measures ensures both computational complexity and differentiability.
3. Comprehensive experiments are conducted to validate the proposal's effectiveness.

**Weaknesses:**

1. Can authors explain why it is important to compare the similarity of representations in graph neural networks?
2. Is NSA sensitive to the removal of low variance principal components from representations?
3. I understand NSA can measure the differences of normalized point-to-point distances in two representations. However, in the definition of Section 3.1, the point structure within representations seems to be not considered. I wonder if there exists a situation where the NSA of two representations is small but the two representations have totally different point structures.
4. Why NSA only slightly outperforms or performs worse than previous methods in some cases in Table 1?

There are also two minor issues:
1. "CKA" in paragraph 1 of Section 2 seems to be in the wrong font.
2. The font size of the text in most figures, especially in Figure 2 is too small.

**Questions:**

See Weaknesses.

---

> ### Author Response · Authors · 2023-11-19
> **Response to Reviewer odB3**
>
> Thank you very much for taking the time to review our work. We address your concerns below:
>
> > Can authors explain why it is important to compare the similarity of representations in graph neural networks?
>
> **Response**: The embedding of complicated metric spaces into simpler spaces has been investigated for a long time. Among the "complicated" metric spaces, graphs have been particular challenging because of impossibility results on isometric embeddings (no matter how many target dimensions) and
> bounds on distortions of their embedding into Euclidean spaces [1][2]. Coupled with their abstraction ability and wide-ranging applications, this makes the study of the geometry of graph representations especially interesting. NSA as a similarity measure is domain agnostic but demonstrating its capability to extract meaningful insights from intricate networks like GNNs will also establish its utility in simpler domains, like images and text processing. We expand upon this in **Concern 1** of the common response and in Section 5 of the revised manuscript.
>
> >  Is NSA sensitive to the removal of low variance principal components from representations?
>
> It is highly likely that as a Representation Similarity Matrix based measure, NSA suffers from the same issues as CKA and it is not sensitive to the removal of low variance/unimportant principal components. We have not conduced experiments to confirm this claim but plan to do so in a future work.
>
> > I understand NSA can measure the differences of normalized point-to-point distances in two representations. However, in the definition of Section 3.1, the point structure within representations seems to be not considered. I wonder if there exists a situation where the NSA of two representations is small but the two representations have totally different point structures.
>
> **Response:** We have expanded on the definition of NSA in Section 3.1 to show how it is capable of measuring point-wise discrepancies along with a global measure of the dissimilarity between two spaces.  We also present new experiments with point-wise NSA in Section 6 (Figure 5 (a)) where we show that NSA can quantify the discrepancy in the relative position of individual points. In our practical tests, we have not encountered a case where the NSA is small but the two representations have completely different point structures. The opposite could be true in the case where the input data lies in a lower manifold dimension compared to its representation dimension. Then the NSA would be large but the two representations (original dimension vs flattened or unfolded manifold dimension) would be very similar. In these situations NSA can be used to minimize the geodesic distance if we know the manifold dimension of the data. We discuss this scenario in Section 4.3 of the paper and present new experiments along with a solution to overcome these situations.
>
> > Why NSA only slightly outperforms or performs worse than previous methods in some cases in Table 1?
>
> **Response:** NSA provides the ideal tradeoff between structure preservation and computational efficiency. NSA is outperformed by PCA in Triplet Accuracy of Cluster Centers. PCA excels at this with its simplicity and focus on linear variance. NSA is also barely outperformed by RTD-AE in RTD as RTD-AE directly minimizes the Representation Topology Divergence between the original and the latent space as part of their training. PCA is strictly a dimensionality reduction technique and cannot be used with mini batching as a loss function. RTD is extremely inefficient in its computational complexity and we present the difference in epoch time for training in Table 3 in Appendix J. RTD-AE takes 10x the time to finish an epoch compared to NSA-AE and is limited to batch sizes under 400 due to its high complexity. We discuss the difference in complexities in Section 3.4 of the revised manuscript. Additionally, the datasets we use for the dimensionality reduction tests are popular benchmark datasets and most existing techniques get high performance when trained on these datasets. We present a more challenging task in Section 4.2 where we train the models to preserve the inter node relationships necessary to perform link prediction. Here we observe NSA-AE outperforming even RTD-AE.
>
> >"CKA" in paragraph 1 of Section 2 seems to be in the wrong font. The font size of the text in most figures, especially in Figure 2 is too small.
>
> We have fixed these errors. Thank you for bringing them to our notice
>
> Please let us know if you have any additional concerns that we can address. If not, would you kindly consider raising your score? Once again, thank you for your review.
>
> [1] On lipschitz embedding of finite metric spaces in Hilbert space by Jean Bourgain, Israel Journal of Mathematics.
> [2] The Geometry of Graphs and Some of Its Algorithmic Applications by Linial, N. and London, E. and Rabinovich, Y. , Proceedings of the 35th Annual Symposium on Foundations of Computer Science

---

> > ### Comment · Reviewer_odB3 · 2023-11-19
> > **Thank you for the response.**
> >
> > Thank you for the response. I have no further questions and I have updated my score.

---

> > > ### Author Response · Authors · 2023-11-19
> > >
> > > Thank you very much for reviewing our revisions and for the updated score. We greatly appreciate your time and the insightful feedback you provided.

---

### Official Review · Reviewer_WXtW · 2023-10-24

**Soundness:** 2 fair
**Presentation:** 3 good
**Contribution:** 2 fair
**Rating:** 3
**Confidence:** 5

**Summary:**

This paper presents 'Normalized Space Alignment' (NSA), a novel
pseudo-metric for comparing representations of high-dimensional
data via their (Euclidean) distances.

Next to mathematically proving relevant properties of a pseudo-metric
like symmetric and the triangle inequality, the paper also presents a
suite of experiments showcasing potential application scenarios. This
includes (a) an analysis of latent representations, (b) analyses of a
set of GNN architectures with respect to adversarial attacks, and (c)
correlation analyses of test accuracy and NSA.

**Strengths:**

- NSA is well-described and all proofs are accessible. The reader is
  guided nicely through the paper (for the most part).

- The paper attempts to cover a broad range of different applications
  for showcasing the utility of NSA.

- The focus on *fundamental* measures is a crucial endeavour for
  improving our understanding of latent representations, quality
  metrics, and much more.

The paper is thus on a good path towards making a strong contribution to
the literature, but, as outlined below, there are some major issues with
the current write-up.

**Weaknesses:**

While I see the contribution favourably, there are major weaknesses
precluding the presentation of the paper in its current form. The
primary one is a **missing analysis of fundamental properties**. While
I appreciate the broad range of different applications, this invariably
means that some depth is lost and analyses are relatively superficial.

Introducing a new measure and then showing its utility requires a more
in-depth view of data, though. For instance, when introducing the
latent representations in Table 1, readers are only shown the actual
scores, but the primary assumption is that the scores actually capture
the relevant properties of the data. Phrased somewhat hyperbolically:
I believe that NSA can be calculated as described, but I do not
understand whether it measures something 'interesting.' The fact that it
correlates with performance metrics is relevant, but immediately
suggests a more detailed comparison scenario, for instance in the form
of an early stopping criterion or regularisation term. Otherwise, most
of the analyses strike me as too speculative. I will provide additional
comments below.

- I'd suggest to shorten the RTD explanation or substantially extend it.
  Currently, it makes use of jargon like $\min G(R, Q)$, 'barcode,' and
  Vietoris--Rips filtrations that are not sufficiently explained (I am
  familiar with the work and I believe that a deep dive into topological
  methods is not required).

- To simplify the notation, I'd either use the actual Euclidean norm in
  the definition of NSA or write $x$ and $0$ as vectors.

- The autoencoders experiment is somewhat out of place since the
  introduction sets up a paper on GNNs. Given the broad scope of NSA,
  I think it might be best to stay with the autoencoder comparison,
  using data with a known ground truth. This could take the form of
  building confidence by starting with simple toy examples like a 'Swiss
  Roll' or other data sets and showing that NSA matches the intuition.

Overall, my **main concern** is that the measure is just too coarse, in
particular given large data sets. It essentially amounts to comparing
averaged distance representations, and more in-depth experiments and/or
theoretical analyses would be required here.

**Questions:**

1. NSA in its current form seems to generalise easily to other distances
   as well. What is the reason for focusing on the Euclidean distance?

2. How robust is the measure and how limited is it in case the 'ambient'
   distances are misleading (such as in the example of a Swiss Roll)?

3. Being based on distances, NSA should be invariant under isometries.
   Is this correct? (I'd overall suggest to simplify the exposition
   here; many of the properties discussed are a direct consequence of
   NSA being based on distances. It is good to be precise and spell that
   out in the appendix, but I'd not give it too much space)

4. How are the representations for Section 3.6 calculated? Is the NSA of
   a specific data set calculated here, i.e. mapping a (batch?) of
   graphs into the latent space? Please clarify!

5. The definition of NSA reminds me of MMD (but lacking the
   cross-comparison term). Could you briefly comment on this?

6. Please show representations of MNIST, F-MNISt, etc.

7. Given the correlation analysis, why not see whether NSA can be used
   to detect or predict a specific level of poisoning?

---

> ### Author Response · Authors · 2023-11-18
> **Response to Reviewer WXtW**
>
> Thank you very much for taking the time to write a detailed review. Your feedback has been extremely useful in updating our work. We address your points of concern below:
>
> >While I see the contribution favourably, there are major weaknesses precluding the presentation of the paper in its current form. The primary one is a missing analysis of fundamental properties. While I appreciate the broad range of different applications, this invariably means that some depth is lost and analyses are relatively superficial.
>
> **Response:** We extend the definition of NSA to show that it can measure point-wise discrepancies. All properties necessary establish its viability as a similarity metric (including its invariance to global isometries) and loss function are proven in Appendix B and C. We also move any speculative results to the appendix (Refer to Summary of Revisions Point 3) and expand on both the autoencoder experiments and the adversarial analysis with more results to cement NSA’s viability as a representation similarity measure. The primary objective of this paper is intended to be the introduction of NSA and a showcase of its versatility. We intend on building on NSA and exploring its potential in-depth in future works.
>
> > Introducing a new measure and then showing its utility requires a more in-depth view of data, though. For instance, when introducing the latent representations in Table 1, readers are only shown the actual scores, but the primary assumption is that the scores actually capture the relevant properties of the data. Phrased somewhat hyperbolically: I believe that NSA can be calculated as described, but I do not understand whether it measures something 'interesting.' The fact that it correlates with performance metrics is relevant, but immediately suggests a more detailed comparison scenario, for instance in the form of an early stopping criterion or regularisation term.
>
> **Response:** Thank you for your feedback. We address this in **Concern 2** and **Concern 3** of the common response. Additionally, we discuss the viability of NSA as an early stopping criterion in Section 5.2 where we show that NSA stabilization correlates with test accuracy convergence.
>
> > I'd suggest to shorten the RTD explanation or substantially extend it. Currently, it makes use of jargon like , 'barcode,' and Vietoris--Rips filtrations that are not sufficiently explained (I am familiar with the work and I believe that a deep dive into topological methods is not required).
> >To simplify the notation, I'd either use the actual Euclidean norm in the definition of NSA or write x and 0 as vectors.
>
> We have taken your feedback into consideration and simplified the explanation of RTD, we have also improved on the notation of NSA in the definition
>
> > The autoencoders experiment is somewhat out of place since the introduction sets up a paper on GNNs. Given the broad scope of NSA, I think it might be best to stay with the autoencoder comparison, using data with a known ground truth. This could take the form of building confidence by starting with simple toy examples like a 'Swiss Roll' or other data sets and showing that NSA matches the intuition.
>
> **Response:** We restructure the paper to reduce the focus on GNNs and address this shortcoming in **Concern 1** of the common response. We expand on the autoencoder experiments and move any speculative results with GNNs to the appendix. We also present the results of using NSA-AE on the Swiss Roll dataset. NSA can be used to minimize geodesic distances if the manifold dimension of the original representation space is known. We exploit this information to flatten a swiss roll with NSA-AE. The results are presented in Section 4.3 and in Figure 1 of the revised manuscript.

---

> > ### Author Response · Authors · 2023-11-18
> >
> > > Overall, my main concern is that the measure is just too coarse, in particular given large data sets. It essentially amounts to comparing averaged distance representations, and more in-depth experiments and/or theoretical analyses would be required here.
> >
> > **Response:** We acknowledge your concern about NSA's perceived coarseness, especially in the context of large datasets. While NSA fundamentally compares averaged distance representations, its design is more nuanced than a simple average. In Section 2 of the revised manuscript, we delve deeper into NSA's structure, demonstrating its capability to discern meaningful pointwise discrepancies in large-scale data. This is further evidenced in our expanded experiments, shown in Figure 5(a), where NSA effectively identifies nodal vulnerabilities in defense mechanisms. Although NSA's simplicity contrasts with more intricate measures like RTD, it is this simplicity that contributes to its computational efficiency and practical applicability.
> >
> > NSA's flexibility also allows for the incorporation of geodesic distances, which can provide deeper insights, particularly for high-dimensional data residing in lower-dimensional manifolds. In cases where the manifold dimension is known, NSA-AE can be trained to minimize the Euclidean distance, effectively approximating the geodesic distances and thus preserving the intrinsic structure of the data. This is demonstrated in our experiments with the Swiss Roll dataset (Figure 1 and Section 4.3), where we compare the outcomes of minimizing both Euclidean and geodesic distances. Additionally, when the manifold dimension is unknown, the latent dimension at which NSA minimization is optimal often reveals the manifold dimension, highlighting NSA's applicability in uncovering the underlying data structure. This approach not only addresses concerns about NSA's coarseness but also underscores its versatility in unfolding complex data structures within an autoencoder framework.
> >
> >
> > >NSA in its current form seems to generalise easily to other distances as well. What is the reason for focusing on the Euclidean distance?
> >
> > **Response:** For NSA to work effectively, it requires a distance measure that is continuous and differentiable. Despite its shortcomings in high dimensional spaces, euclidean distance is a popular measure and is proven to be an effective alignment objective in several domains. We also discuss this in *Paragraph 3 of the Discussion and Conclusion section (Section 7)*. A potential improvement on euclidean distances would be geodesic distances. But geodesic distance preservation is not continuous when it is used as a minimization objective. We are currently working on developing a differentiable approximation of geodesic distance which would be better suited for alignment tasks in high dimensional spaces, but euclidean distance works well in most training paradigms.
> >
> > > How robust is the measure and how limited is it in case the 'ambient' distances are misleading (such as in the example of a Swiss Roll)?
> >
> > **Response:** Euclidean distance as an alignment objective is sufficiently robust to outliers during training due to our normalization technique (Section 3.5). But it can fail to accurately capture relative distance when the data forms complex structures where ‘ambient’ distances are misleading. We discuss this in more detail and present the results of the Swiss Roll experiment in Section 4.3 and Figure 1. If the manifold dimension of the original data is known, euclidean distance between the points in the manifold space would be an approximation of geodesic distances in the original space. Thus the geodesic distances can be minimized by minimizing the euclidean distance using NSA, which results in the representation structure being unfolded to its manifold dimension.
> >
> > >Being based on distances, NSA should be invariant under isometries. Is this correct?
> >
> > **Response:** Yes you are correct, a similarity measure like NSA is invariant under any global structure preserving isometry such as scaling, rotation or translation. But it is not invariant to local transformations that preserve functional similarity but not structural similarity such as subset translations. We provide proofs that NSA satisfies the above isometries in Appendix B.

---

> > > ### Author Response · Authors · 2023-11-18
> > >
> > > > How are the representations for Section 3.6 calculated? Is the NSA of a specific data set calculated here, i.e. mapping a (batch?) of graphs into the latent space? Please clarify!
> > >
> > > **Response:** To compute the similarity between layers in Section 5.1 (3.6 pre-revision), we utilize the intermediate embeddings (final output embeddings for the final layer) of the input data from each layer and use these to compute layerwise similarity. For each NSA computation, we sample 4000-9000 points (randomly sampled for different runs) from the input data and obtain their corresponding intermediate embeddings across both models. NSA is then computed on this sample. Similarly for RTD, we sample 400 points (the recommended sample size by the paper) and compute the RTD on this sample. We have clarified this in Section 5.1
> > >
> > > > The definition of NSA reminds me of MMD (but lacking the cross-comparison term). Could you briefly comment on this?
> > >
> > > **Response:** MMD is fundamentally different from NSA because it measures the global discrepancy between two distributions. It does this by calculating a global term for the first and the second distribution individually. This means that MMD cannot preserve pairwise discrepancies and a global term might result in two opposing values canceling out (which should not happen as both represent a discrepancy of equal magnitude). NSA also needs to preserve pointwise identities which MMD cannot because it has global terms.
> > >
> > > > Please show representations of MNIST, F-MNIST, etc.
> > >
> > > We show the visualization of representations obtained from MNIST, F-MNIST and COIL-20 in Appendix L. T-SNE is used to reduce the latent dimension of the embeddings to 3.
> > >
> > > > Given the correlation analysis, why not see whether NSA can be used to detect or predict a specific level of poisoning?
> > >
> > > **Response:** The correlation between NSA and network performance metrics, particularly under poisoning scenarios, indeed suggests potential for NSA in detecting or predicting specific levels of poisoning. While we did not explicitly explore this application in the current study, it represents a compelling direction for future research. To effectively implement this, we would first establish a baseline correlation between NSA values and known poisoning levels in controlled environments. Subsequently, we could develop a predictive model leveraging these correlations to estimate poisoning levels in unknown datasets. This is a good direction and we plan on implementing this in a future work, where NSA's application in detecting and preventing adversarial attacks and data poisoning will be explored.
> > >
> > >
> > > If we have have effectively addressed your concerns, would you kindly consider increasing your score? If not, could you let us know of any additional changes that you would like to see?

---

> > > > ### Comment · Reviewer_WXtW · 2023-11-20
> > > >
> > > > I appreciate the changes and I believe that they are going in the right direction. Nevertheless, my main concerns remain unaddressed. There is still no discussion on the properties of the proposed measure with respect to capturing fundamental characteristics of data. I understand that NSA is shown to be a pseudo-metric, but there are numerous ways of defining pseudo-metrics between spaces; the property of being a pseudo-metric does not necessarily imply that salient characteristics are captured.
> > > >
> > > > All the evidence for the practical utility of NSA is empirical so far, but introducing a novel measure like this requires a more in-depth analysis. I believe that such an analysis can be provided but it would involve a **major revision of the paper**, which is beyond the scope of this conference cycle.

---

> > > > > ### Author Response · Authors · 2023-11-23
> > > > >
> > > > > Thank you for your valuable feedback regarding the exploration of NSA's fundamental properties. We recognize the importance of theoretical underpinnings in representation analysis techniques. However, it is crucial to note that NSA serves a dual purpose, distinguishing it from techniques like CKA and RTD. While these existing methods are primarily geared towards analysis, NSA is uniquely positioned as both a loss function and an analysis tool.
> > > > >
> > > > > This dual functionality of NSA has directed the focus of our current work. Unlike methods that are purely analytical and thus require extensive theoretical grounding to validate their efficacy, NSA's practical viability as a loss function can be demonstrated through a broad range of empirical scenarios. Our paper, therefore, concentrates on showcasing NSA's effectiveness and efficiency across diverse applications, highlighting its adaptability and potential as a tool for minimizing representation structure discrepancies.
> > > > >
> > > > > A deeper theoretical exploration of NSA would add significant value to understanding its performance, and we have included some foundational theory to this effect. We hope that the reviewer can appreciate the unique positioning of NSA and evaluate our work within the context of its intended application scope and the promise it holds for future theoretical and empirical developments.
> > > > >
> > > > > We have made minor phrasing changes throughout the paper to better represent the unique position of NSA in this research domain. We also provide additional theory and empirical results in Appendix D to prove why NSA is currently the only viable structure discrepancy minimization term.

---

### Official Review · Reviewer_wTyL · 2023-10-30

**Soundness:** 2 fair
**Presentation:** 1 poor
**Contribution:** 1 poor
**Rating:** 3
**Confidence:** 2

**Summary:**

The paper proposes a new pseudometric as a way to measure the distance between two representation spaces. The authors then show how it can be used in various applications.

**Strengths:**

- The authors tackle a significant problem, with many downstream applications
- The metric is reasonably novel

**Weaknesses:**

- The proposed pseudometric does not make any sense to me. One clear issue with it is the fact that it is not invariant to permutation between the vectors in the point cloud. This is a huge issue, in my mind, as there is no reason to assume the vectors are aligned so it is greatly impacted by a random permutation between them. The authors also don't give any sufficient motivation for this metric, besides the fact that it satisfies some basic properties like triangle inequality.
- The paper is not well written and very confusing to read. Comparing representations and the proposed method is not specific to GNNs, yet the authors present it as one that is connected to GNNs. They also move between GNNs and point clouds making it hard to follow
- The use of NSA as a regularizer in the VAE part is not clear at all. Do you compare the reconstructed with the original? It was not stated clearly. If this is the case, then this does not prove the usefulness of the NSA as the representations are aligned.

**Questions:**

What is the motivation behind the NSA definition?

---

> ### Author Response · Authors · 2023-11-19
> **Response to Reviewer wTyL**
>
> Thank you for taking the time to review our work. We address all your concerns below:
>
> > The proposed pseudometric does not make any sense to me. One clear issue with it is the fact that it is not invariant to permutation between the vectors in the point cloud. This is a huge issue, in my mind, as there is no reason to assume the vectors are aligned so it is greatly impacted by a random permutation between them. The authors also don't give any sufficient motivation for this metric, besides the fact that it satisfies some basic properties like triangle inequality.
>
> **Response:** Thank you for taking the time to review our work. NSA is a representation similarity measure and measures the discrepancy between two representation spaces. A permutation between the vectors in a point cloud would result in a loss of the one-to-one mapping that NSA requires and would fundamentally change the representation space and the relationships between the points of this space, thus representation similarity measures like NSA or CKA cannot be invariant to this type of permutation. NSA as a similarity measure is invariant to global isometries.
>
> NSA along with other Representation Similarity measures [1] are invariant to scaling, translation or rotation transformations on the representation spaces as they only compare the discrepancy between the relative positions of data in their own embedding spaces. Representation Similarity Measures are well motivated through several popular previous works like CCA [2], CKA [3], RSA [4] and RTD [5]. We provide proofs in Appendix A that NSA satisfies all the conditions to qualify as a pseudometric. Appendix B provides the proofs to demonstrate that it is invariant to isometries and satisfies the conditions necessary to qualify as a similarity index, as proposed by [3]. Appendix C proves that NSA is continuous and differentiable, allowing us to use it as a minimization objective.
>
> NSA is motivated by the need to fill a gap in existing similarity measure research. Current measures fall into one of the two categories: 1) They can quantify the discrepancy between two representation spaces but are not differentiable and cannot be used as a minimization objective in neural network training. 2) Measures that are differentiable [5] are very computationally expensive and cannot be employed in practical scenarios for large scale neural network training. NSA is differentiable and its complexity is quadratic in the number of points, facilitating its use as a minimization objective in any training paradigm that necessitates the alignment of embedding spaces such as dimensionality reduction, adversarial training, link prediction, knowledge transfer or distillation and semantic similarity among others without a significant loss in training efficiency. We present results that show NSA not only preserves representation structure (Section 4) but the aligned embeddings are effective in downstream tasks like link prediction (Section 4.2) and semantic similarity (Appendix M). We also show in Table 3 that NSA is over 10x faster than its closest competitor (RTD) while being used as an additional loss term. All of these experiments serve to showcase the versatility of NSA and its ability to serve in practical training scenarios.
>
>
>
> [1] Similarity of Neural Network Models: A Survey of Functional and Representational Measures by Klabunde, Max and Schumacher, Tobias and Strohmaier, Markus and Lemmerich, Florian, arXiv 2023.
> [2] SVCCA: Singular Vector Canonical Correlation Analysis for Deep Learning Dynamics and Interpretability by Raghu, Maithra and Gilmer, Justin and Yosinski, Jason and Sohl-Dickstein, Jascha NeurIPS 2017.
> [3] Similarity of Neural Network Representations Revisited by Simon Kornblith, Mohammad Norouzi, Honglak Lee and Geoffrey E. Hinton.
> [4] Representational similarity analysis - connecting the branches of systems neuroscience by Kriegeskorte, Nikolaus and Mur, Marieke and Bandettini, Peter, Frontiers in Systems Neuroscience 2008.
> [5] Representation Topology Divergence: A Method for Comparing Neural Network Representations by Barannikov, Serguei and Trofimov, Ilya and Balabin, Nikita and Burnaev, Evgeny. ICML 2022.

---

> > ### Author Response · Authors · 2023-11-19
> >
> > > The paper is not well written and very confusing to read. Comparing representations and the proposed method is not specific to GNNs, yet the authors present it as one that is connected to GNNs. They also move between GNNs and point clouds making it hard to follow
> >
> > **Response:** We address your concern related to the structuring of the paper and its focus on GNNs in **Concern 1** of the common response. We have fixed the writing to present NSA as a domain agnostic measure and restructured the paper so that it focuses on introducing NSA as a similarity measure first, proving its viability as a loss function next and then ending with any additional empirical analysis. GNNs are the architectures that we focus our empirical analysis on, we explain our reasoning for this in Section 5 of the manuscript. Point clouds refer to sets of data points in a high-dimensional space which are usually the output embeddings of neural networks. They lie in a high dimensional representation space and NSA measures the discrepancy between two point clouds. Please let us know if there are any particular aspects we can improve upon to help with the readability.
> >
> > > The use of NSA as a regularizer in the VAE part is not clear at all. Do you compare the reconstructed with the original? It was not stated clearly. If this is the case, then this does not prove the usefulness of the NSA as the representations are aligned.
> >
> > **Response**: Thank you for your question. I think there is a misunderstanding on how NSA is used in autoencoders. NSA is not used as a regularizer for VAEs but as an additional loss term for a regular autoencoder. We do not present any experiments on using NSA with VAEs but show that NSA can help improve the latent representations obtained from a regular autoencoder. Furthermore, NSA acts directly on the outputs of the latent space and not on the weights, acting as a loss term and not as a regularizer.
> >
> > A normal autoencoder aims to minimize the MSE loss between the original embedding $X$ and reconstructed embedding $\hat{X}$. We add NSA Loss as an additional loss term that aims to minimize the discrepancy in representation structure between the original embedding space $X$ and the latent embedding space $Z$. The autoencoder is built as a compression autoencoder where the encoder attempts to reduce the original data to a latent dimension and the decoder attempts to re-construct the original embedding. The latent embeddings $Z$ are extracted from the final layer of the encoder. Thus in an NSA enhanced autoencoder, there are two loss functions acting in tandem:
> > 1) The MSE loss attempts to minimize the reconstruction loss between the original embedding $X$ and the reconstructed embedding $\hat{X}$
> > 2) The NSA loss attempts to minimize the NSA between the original embedding $X$ and the latent embedding $Z$.
> >
> > We have improved the writing in Section 4 to make this more explicit. Please let us know if there is anything else that we can expand upon or clarify for you to help clear any concerns that you might have with the paper. You can also refer to the common response where we present a list of all the major revisions made to the manuscript during the rebuttal period and address common concerns regarding the manuscript. We discuss NSA's motivation in more depth in **Concern 1** and **Concern 4** specifically. We address the additional experiments we have included in our revision in **Concern 2** and **Concern 3**. If we have resolved all your concerns, we hope you will kindly consider raising your score for our paper.

---

### Official Review · Reviewer_BYXr · 2023-10-31

**Soundness:** 2 fair
**Presentation:** 2 fair
**Contribution:** 2 fair
**Rating:** 3
**Confidence:** 4

**Summary:**

This manuscript presents a novel approach for comparing two point clouds through the introduction of a Normalized Space Alignment (NSA) technique, aimed at assessing the pairwise distances between corresponding points as a means of quantifying the similarity between two representations. The authors show that NSA possesses the properties of a pseudometric and demonstrate its applicability as a loss function within the context of autoencoder models. A set of experimental studies is conducted, offering insights into the behavior and performance of various graph neural network representations when analyzed using this newly proposed method.

**Strengths:**

1. The authors have introduced a Normalized Space Alignment (NSA) technique, combining the computational efficiency of Centered Kernel Alignment (CKA) with the differentiability of Representation Tree Distance (RTD).

2. The manuscript provides a theoretical demonstration, showcasing that NSA exhibits the properties of a pseudometric. Additionally, the authors have developed a differentiation scheme, enabling the integration of NSA as a loss function.

**Weaknesses:**

Concerns on Structure Preservation:

   1.The authors posit that the Normalized Space Alignment (NSA) technique is structure-preserving, yet there is a noticeable gap in addressing the underlying data's graph structure. NSA primarily focuses on measuring distances between two point clouds, neglecting the crucial graph edges. This oversight brings its capability to preserve graph structure into question. Moreover, although the paper intends to underscore NSA's unique potential in the context of Graph Neural Networks, the actual exploration of these capabilities is missing.

Issues with Experimental Validation:
 1. Lack of Downstream Task Evaluation: Given that NSA is proposed as a loss function, it is critical to evaluate its effectiveness in downstream tasks. Unfortunately, the paper lacks such evaluations. Insights into how the latent embeddings from an autoencoder, shaped by NSA, could enhance downstream task performance are notably missing.

  2. Dataset Limitations: The paper’s conclusions are drawn from experiments conducted solely on the Amazon Computers dataset for node classification. Different graph datasets possess varied properties and can elicit diverse behaviors, making it imperative to extend the analysis to a broader set of datasets for more robust and convincing results.

  3. Inconsistency in Data Analysis: There is a puzzling contrast between the paper’s stated focus on Graph Neural Networks (and graph data) and the NSA-AE analysis, which is not applied to graph datasets.

   4. Adversarial Attack Analysis Shortcomings: The attempt to correlate NSA values with misclassification rates, aiming to use NSA as a metric for evaluating GNN resilience, lacks conviction. NSA values exhibit significant variability across different GNN architectures, and potentially across various graph datasets. The observed discrepancy, where GCN shows the highest misclassification rate while GCN-SVD has the highest NSA value, further complicates any straightforward interpretation based on NSA values alone.

  5. Readability of Figures: The figures included in the paper suffer from readability issues, with some fonts being excessively small, hindering the reader’s ability to fully grasp and interpret the presented data.

**Questions:**

I have raised several points and posed various questions in the previous sections of my review.

Additionally, I have a specific inquiry pertaining to Figure 3. Regarding Figure 3, could you please clarify which two representations are being compared to calculate the NSA values presented? The text does not seem to provide explicit information on this aspect.

---

> ### Author Response · Authors · 2023-11-18
> **Response to Reviewer BYXr**
>
> Thank you very much for taking the time to review our work. We will address your concerns below:
>
> >The authors posit that the Normalized Space Alignment (NSA) technique is structure-preserving, yet there is a noticeable gap in addressing the underlying data's graph structure. NSA primarily focuses on measuring distances between two point clouds, neglecting the crucial graph edges. This oversight brings its capability to preserve graph structure into question. Moreover, although the paper intends to underscore NSA's unique potential in the context of Graph Neural Networks, the actual exploration of these capabilities is missing.
>
> NSA belongs to the class of similarity indices just like Centered kernel alignment and Representation Topology Divergence. NSA measures the discrepancy between two embedding spaces. As a similarity index it is capable only of measuring how dissimilar two embedding spaces are. As a minimization objective, it is capable of aligning one embedding space to another (parent or ideal) embedding space. Scenarios where embedding space alignment is of paramount importance would be knowledge transfer, distillation, dimensionality reduction, robust training and model interpretability among others. We address the issue of the paper being too focused on GNNs in **Concern 1** of the common response. Another reason for the paper utilizing GNNs in its empirical analysis is that exploration of GNN’s embedding capabilities are largely missing from previous literature.
>
>
>
>
> > Lack of Downstream Task Evaluation: Given that NSA is proposed as a loss function, it is critical to evaluate its effectiveness in downstream tasks. Unfortunately, the paper lacks such evaluations. Insights into how the latent embeddings from an autoencoder, shaped by NSA, could enhance downstream task performance are notably missing.
>
> We present additional experiments to validate NSA-AE’s downstream task performance in section 4.2 and appendix M. Please refer to **Concern 2** of the common response for additional details.
>
>
>
> > Dataset Limitations: The paper’s conclusions are drawn from experiments conducted solely on the Amazon Computers dataset for node classification. Different graph datasets possess varied properties and can elicit diverse behaviors, making it imperative to extend the analysis to a broader set of datasets for more robust and convincing results.
>
> We choose to train our models on the Amazon Computer Dataset as it has the ideal tradeoff between downstream task accuracy and dataset size. GNNs are almost always shallow as the benchmark datasets currently available and utilized by the research community are very small. Training a 2 layer GNN on these datasets would not show a strong linear correlation across layers that we wanted to demonstrate with NSA. The larger datasets have very poor performance with popular GNN architectures, often not reaching convergence. Since all our empirical analysis relies on the assumption that representation spaces stabilize as they converge (Figure 3), a poor performing dataset cannot be used for our experiments.
>
> We extend our experiments to 4 additional datasets to demonstrate NSA’s robustness. We utilize the Cora, Citeseer, Pubmed and Flickr datasets and showcase their results on the sanity tests, cross architecture tests and cross downstream task tests in Appendix Q.
>
>
>
> > Inconsistency in Data Analysis: There is a puzzling contrast between the paper’s stated focus on Graph Neural Networks (and graph data) and the NSA-AE analysis, which is not applied to graph datasets.
>
> Thank you for bringing this shortcoming to our attention. We address this in **Concern 1** of the common response and have made the necessary revisions in our manuscript to improve the flow of the content in the paper.
>
>
>
> > Adversarial Attack Analysis Shortcomings: The attempt to correlate NSA values with misclassification rates, aiming to use NSA as a metric for evaluating GNN resilience, lacks conviction. NSA values exhibit significant variability across different GNN architectures, and potentially across various graph datasets. The observed discrepancy, where GCN shows the highest misclassification rate while GCN-SVD has the highest NSA value, further complicates any straightforward interpretation based on NSA values alone.
>
>  Thank you for your feedback. We address this in **Concern 3** of the common response and present additional experiments in Section 6 to help interpret the initial results better. We have also improved the writing in this section to help with readability.

---

> > ### Author Response · Authors · 2023-11-18
> >
> > > Readability of Figures: The figures included in the paper suffer from readability issues, with some fonts being excessively small, hindering the reader’s ability to fully grasp and interpret the presented data.
> >
> > We have increased the font size of most of the figures in our paper. Any further increase in font sizes will lead to issues with the structure and formatting of the figures. Please let us know if any figure specifically still has this issue and we will fix it.
> >
> > > Additionally, I have a specific inquiry pertaining to Figure 3. Regarding Figure 3, could you please clarify which two representations are being compared to calculate the NSA values presented? The text does not seem to provide explicit information on this aspect.
> >
> > In Figure 4 (post revision) we showcase NSA across varying perturbation budgets in an adversarial attack.
> > For an evasion attack we utilize a model trained on a clean graph. The first embedding space is obtained from inferring a clean graph on this model, the second embedding space is obtained from inferring a perturbed graph (with the corresponding perturbation budget) on the same model.
> > For a poisoning attack, we train a model on a clean graph and infer embeddings from this clean graph. We train another model with the same architecture on a perturbed graph and infer embeddings from the perturbed graph on this model. These two embeddings are now compared and the discrepancy between the clean embedding and the perturbed embedding is quantified by NSA in Figure 4.
> > We have improved the writing in this section to clarify this information.
> >
> >
> > Please let us know if you have any other questions. If we have addressed your concerns would you kindly consider improving your score?

---

### Author Response · Authors · 2023-11-18
**Global Response addressing common concerns**

We would like to extend our sincere gratitude to all the reviewers for their insightful comments and constructive criticisms. Your feedback has been invaluable in enhancing the clarity, depth, and rigor of our research. The following response aims to address common concerns across the reviewers and present a summary of the revisions made to our manuscript. We will address reviewer specific concerns individually.

**Summary of Major Revisions to the Manuscript**
1. The Introduction was re-written to better focus on NSA as a domain agnostic representation similarity measure.
2. Restructured the paper to focus on NSA’s definition and its application as a loss function first. Moved empirical analysis to the end. Section 3.6 (Comparing different initialization) and 3.7 (Convergence Tests) are now Section 5.1 and 5.2.
3. Moved Section 6 (Are GNNs task specific learners) to the Appendix (Appendix F)
4. Shortened Related Work to remove technical jargon from CKA and RTD that does not get expanded upon.
5. Expanded on the definition of NSA. NSA can now measure pointwise discrepancies and globally measure the discrepancy between representation spaces. Revised definition is presented in Section 3.1.
6. Moved NSA’s differentiation and continuity proofs to the Appendix (Appendix C).
7. Increased the font size and removed unnecessary information from all the figures.
8. Improved the writing in Section 4.1 (Defining NSA-AE) to help better understand how NSA works as a loss function in autoencoders.
9. Section 4.2 was added which presents the results of downstream task analysis on the latent embeddings obtained from NSA-AE. Table 2 was added which presents the results of experiments on Link Prediction. Appendix M presents the results of experiments on Semantic Text Similarity.
10. Section 4.3 was added which demonstrates how NSA-AE can utilize geodesic distances to unfold complex representation structures into their lower dimensional manifold representation. Figure 1 was added which shows the results of these experiments on the Swiss Roll dataset.
11. The empirical analysis section (Section 5) was expanded upon to explain why we work with Graph Neural Networks and better explain how the sanity test is conducted (Section 5.1)
12. The convergence test section (Section 5.2) and figures (Figure 3) now show the point at which NSA converges. This point can be utilized during training as an early stopping objective and its correspondence with test accuracy is shown.
13. The adversarial attack section (Section 6) is expanded upon to better explain how NSA is measured for the models across perturbations. Speculative results are removed and additional experimental results are presented in Figure 5 as additional analyses to support the initial results obtained.
14. The Discussion and conclusion section (Section 7) is condensed to remove speculative arguments.
15. Additional figures were added in Appendix L (Figure 18) to show the results of visualizing the latent embeddings obtained from the autoencoders.
16. Additional experimental results were added in Appendix Q showcasing the results of empirical analyses involving sanity tests, cross architecture tests and cross downstream task tests on 4 additional graph datasets.

---

> ### Author Response · Authors · 2023-11-18
>
> **Concern 1: Why is NSA focused on Graph Neural Networks? The switch from GNNs to autoencoders seems out of place. Why use GNNs to evaluate NSA if you do not explore the correlation with graph structure?**
>
> **Response:** We appreciate your queries regarding our focus on Graph Neural Networks (GNNs) and the transition to autoencoders in our study. Initially, we chose GNNs due to the representation challenges in embedding graph structures into simpler spaces, a topic of significant interest in recent research. Graphs have been particularly challenging because of impossibility results on isometric embeddings (no matter how many target dimensions) and bounds on distortions of their embedding into Euclidean spaces [1][2].  GNNs, with their inherent complexity, provided a robust testing ground for NSA. NSA is designed to measure 'dissimilarity' between two embedding spaces as long as a one to one mapping exists between the points in both embedding spaces, regardless of their originating architecture or dimensionality. This flexibility allows NSA to be applicable across various domains, and not only limited to GNNs.
>
> Our initial focus on GNNs aimed to demonstrate NSA's capability to extract meaningful insights from intricate neural networks, thus establishing its utility in simpler domains, like image or text processing. Previous literature on similarity indices is also focused on convolutional neural networks and regular neural networks, leaving GNN embedding capabilities largely unexplored. However, we acknowledge that the paper's initial version may have overly emphasized GNNs, potentially obscuring NSA's broader applicability.
>
> In response to your feedback, we have revised the manuscript to clarify NSA's versatility. We have restructured the content to introduce NSA as a domain-agnostic similarity metric initially, followed by its validation through autoencoder experiments. The latter part of the paper now focuses on empirically analyzing NSA's effectiveness, using GNNs as a case study to illustrate its performance rather than as the primary subject of investigation. These changes aim to present a more balanced view, highlighting NSA's general applicability rather than suggesting it is specific to GNNs
>
>
> **Concern 2: Autoencoder results are not convincing enough. A Downstream task evaluation of the embeddings obtained from NSA would help prove its viability.**
>
> **Response:** We concur that maintaining structural consistency between the original and latent embedding spaces in our autoencoder doesn't intrinsically validate the utility of the latent embeddings. Specifically, in tasks where preserving semantic relationships and the relative positions of data points is critical, traditional dimensionality reduction might not suffice. This concern is particularly relevant in applications like semantic text similarity search, link prediction, facial recognition, and knowledge transfer tasks.
>
> To address this, we have extended our evaluation to include experiments where the NSA-AE's performance is tested in practical scenarios. We trained NSA-AE using embeddings from a link prediction model, then assessed the latent embeddings' efficacy in preserving global structural information. The results, presented in Table 2 of the main text, show NSA-AE outperforming both a basic autoencoder and RTD-AE in retaining this crucial information.
>
> Furthermore, we evaluated NSA-AE on semantic textual similarity tasks. Here, we reduced word embeddings from a word2vec model and compared the correlation between similarities derived from the original and latent embeddings using the Semantic Textual Similarity Multi EN dataset. These findings, detailed in Appendix M, further substantiate the practical utility of NSA-AE's latent embeddings.
>
> The results that we present in the revised manuscript effectively address your concerns regarding the utility of embeddings obtained from NSA, particularly in the contexts we tested. Our intention was to demonstrate NSA’s applicability and effectiveness in specific scenarios, like semantic textual similarity and link prediction tasks. Looking forward, we recognize the potential of exploring NSA’s utility in broader domains, such as knowledge distillation and adversarial training. However, these areas represent a substantial expansion of our current scope and necessitate significant additional research. We are keen on undertaking this endeavor and plan to present our findings in future work. Our goal is to continually expand the understanding and applicability of NSA, and we value the guidance your feedback has provided in steering our research in this direction.
>
>
> [1] On lipschitz embedding of finite metric spaces in Hilbert space by Jean Bourgain, Israel Journal of Mathematics.
> [2] The Geometry of Graphs and Some of Its Algorithmic Applications by Linial, N. and London, E. and Rabinovich, Y. , Proceedings of the 35th Annual Symposium on Foundations of
> Computer Science

---

> > ### Author Response · Authors · 2023-11-18
> >
> > **Concern 3: Results of the adversarial analysis are speculative.**
> >
> > **Response:** Thank you for pointing out the need for a more rigorous foundation in our analysis of adversarial attacks, particularly regarding SVD-GCN. Our initial intention was to highlight the significance of preserving representation structure in embedding spaces as a measure of robustness against adversarial attacks. In doing so, we referenced work by [1] where adaptive attacks demonstrated SVD-GCN's vulnerability, contradicting its previously assumed strength based on high classification accuracy against common attack methods.
> >
> > In our study, while NSA values generally correlated with misclassification rates under poisoning attacks, the results under evasion attacks presented an interesting deviation. Notably, SVD-GCN exhibited the highest NSA values, indicating a substantial alteration in the embedding space post-attack, even at varied perturbation rates. NSA, in this context, was calculated by comparing the embedding spaces from a clean graph versus those inferred from a perturbed graph under evasion attacks.
> >
> > Upon reflection, we acknowledge that our initial interpretation linking SVD-GCN's high NSA values directly to its failure under adaptive attacks required further analysis. To address this, we have included additional evidence in Figure 5 of our revised manuscript. Here, we analyze the 'boundary nodes' in the representation spaces of various GNN architectures. Boundary nodes, characterized by low classification confidence and proximity to the decision boundary, significantly increased in number in SVD-GCN post-attack. This increase, accompanied by a notable decline in classification confidence across all nodes, could explain the observed high NSA values. Furthermore, we assess the pointwise NSA of these boundary nodes, finding that SVD-GCN has the highest values among the tested architectures. This analysis suggests a potential explanation for SVD-GCN's high NSA values and its vulnerability.
> >
> > Our aim was to demonstrate that a simple empirical analysis using NSA could reveal hidden vulnerabilities in defense methods, which might not be apparent through traditional metrics alone. This additional analysis provides a more solid foundation for our findings and addresses the concerns raised in your feedback.
> >
> > **Concern 4:** NSA lacks motivation and is coarse.
> >
> > **Response:** We appreciate your feedback regarding the motivation and perceived coarseness of the Normalized Space Alignment (NSA). NSA, as part of the representation similarity matrix (RSM) based measures, continues a well-established tradition in similarity analysis, with methods like CKA demonstrating their effectiveness over the years. Our work with NSA is driven by a specific goal: to provide a computationally efficient and differentiable similarity measure, addressing a gap in existing research.
> >
> > The uniqueness of NSA lies in its combination of computational efficiency and differentiability, features not commonly found together in existing similarity measures. Most current measures, due to their normalization schemes or methods of quantifying similarity, lose the capacity to be differentiable and, thus, cannot be directly used as objective functions to minimize dissimilarity. For example, the recently introduced RTD metric, while differentiable, is cubic in the number of simplices involved, which might be significantly higher than the number of points, rendering it inefficient for practical applications. NSA, in contrast, is designed to be quadratic in complexity and maintains differentiability, positioning it as a valuable tool for a range of applications.
> >
> > Moreover, NSA is not limited to providing a global value that quantifies the discrepancy between two spaces. It can also be applied as a pointwise measure, assessing the discrepancy for individual points across spaces. We detail this aspect of NSA by expanding on its definition in Section [specific section]. We also demonstrate this by including its application in identifying local vulnerabilities within an embedding space. Particularly, in Figure 5(a) we show that we can measure the pointwise NSA of boundary nodes during an evasion attack and help in pinpointing vulnerable nodes, enabling targeted analyses to address their weaknesses.
> >
> > The primary aim of this paper is to introduce NSA as a novel metric filling a previously unaddressed need in similarity indices. Future work will focus on providing a comprehensive theoretical basis for NSA, along with exploring its application in domain-specific tasks such as knowledge distillation and adversarial training in neural networks. We believe that the current work lays a solid groundwork for NSA's potential and utility in these areas.
> >
> >
> > [1] Are Defenses for Graph Neural Networks Robust? by Mujkanovic, Felix and Geisler, Simon and Gunnemann, Stephan and Bojchevski, Aleksandar, NeurIPS 2022

---

> > > ### Comment · Reviewer_WXtW · 2023-11-20
> > >
> > > > Concern 4: NSA lacks motivation and is coarse.
> > >
> > > The response by the authors already mentions relevant points. I need to point out that the main criticisms of my fellow reviewer colleagues and myself points out that, in contrast to metrics like RTD or CKA, the current version of the manuscript misses a more detailed analysis of what type of properties are actually captured by NSA. RTD, for example, can be linked to topological features, and thus benefits from a wealth of additional stability theorems, estimates, and so on. Additional analyses would really strengthen the paper here.

---

### Author Response · Authors · 2023-11-23
**Final set of revisions to manuscript**

We have made the following additional changes to our manuscript based on the reviewers’ feedback:

1. Throughout the paper, we have revised the phrasing to emphasize NSA's dual role as both a manifold comparison technique and a differentiable loss function. This alteration enhances the clarity of NSA's unique position in the field of representation space quantification.

2. New Appendix (Appendix D): We have added a section where we delve into NSA's compatibility with mini-batching, crucial for its application as a loss term. This includes: (a.) **Theoretical Foundation:** Lemma 10 provides a theoretical basis, proving that, in expectation, NSA calculated over a mini-batch equates to the NSA of the entire dataset. (b.) **Empirical Validation:** We present empirical evidence showing that NSA, over numerous trials, approximates the NSA of the entire dataset. This is in contrast to RTD, which fails to demonstrate similar consistency.

We hope these enhancements provide a better perspective of NSA's capabilities.

---

### Author Response · Authors · 2023-11-23
**Request for Review of Revisions and Responses Before Rebuttal Period Closure**

Dear Reviewers,

As the rebuttal period is nearing its end, we would like to respectfully request you to review the revisions and responses we have submitted for our manuscript. Your feedback is invaluable to us, and we have made significant changes to the paper based on the comments received thus far.

We understand the time constraints and appreciate the effort you put into reviewing papers. Your insights are crucial for the final version of our manuscript, and we hope to have addressed all the concerns raised.

Thank you for your time and consideration.

---

### Meta-Review · Area_Chair_4ND8 · 2023-12-05

**Metareview:**

The paper is not well written and most reviewers thinks it is hard to read and follow, and they have lots of concerns about the technical details. The authors are encouraged to carefully polish their paper when preparing the next venue.

**Justification For Why Not Higher Score:**

N/A

**Justification For Why Not Lower Score:**

N/A

---

### Decision · Program_Chairs · 2024-01-16

Reject